# Dual-View Predictive Diffusion: Lightweight Speech Enhancement via Spectrogram-Image Synergy

**Ke Xue** [1]  **Rongfei Fan** [1]  **Kai Li** [2]  **Shanping Yu** [1]  **Puning Zhao** [3]  **Jianping An** [1]

## Abstract

Diffusion models have recently set new benchmarks in Speech Enhancement (SE). However, most existing score-based models treat speech spectrograms merely as generic 2D images, applying uniform processing that ignores the intrinsic structural sparsity of audio, which results in inefficient spectral representation and prohibitive computational complexity. To bridge this gap, we propose **DVPD**, an extremely lightweight **D**ual-**V**iew **P**redictive **D**iffusion model, which uniquely exploits the dual nature of spectrograms as both visual textures and physical frequency-domain representations across both training and inference stages. Specifically, during training, we optimize spectral utilization via the Frequency-Adaptive Non-uniform Compression (FANC) encoder, which preserves critical low-frequency harmonics while pruning high-frequency redundancies. Simultaneously, we introduce a Lightweight Image-based Spectro-Awareness (LISA) module to capture features from a visual perspective with minimal overhead. During inference, we propose a Training-free Lossless Boost (TLB) strategy that leverages the same dual-view priors to refine generation quality without any additional fine-tuning. Extensive experiments across various benchmarks demonstrate that DVPD achieves state-of-the-art performance while requiring only **35%** of the parameters and **40%** of the inference MACs compared to SOTA lightweight model, PGUSE. These results highlight DVPD's superior ability to balance high-fidelity speech quality with extreme architectural efficiency. Code and audio sam-

ples are available at `https://github.com/ke12345213/dvpd_demo`

[1]School of Cyberspace Science and Technology, Beijing Institute of Technology, Beijing 100081, China [2]Department of Computer Science and Technology, Institute for AI, BNRist, Tsinghua University, Beijing 100084, China [3]School of Cyber Science and Technology, Sun Yat-Sen University, Shenzhen 518107, China. Correspondence to: Rongfei Fan <fanrongfei@bit.edu.cn>.

*Proceedings of the 43$^{rd}$ International Conference on Machine Learning*, Seoul, South Korea. PMLR 306, 2026. Copyright 2026 by the author(s).

## 1. Introduction

In real-world acoustic environments, speech signals are commonly degraded by noise (Braun et al., 2021), reverberation, clipping (Mack & Habets, 2019), and other distortions, which severely impair speech quality and intelligibility. Speech enhancement (SE) seeks to recover high-fidelity speech from corrupted signals. Existing SE methods (Braun et al., 2021; Mack & Habets, 2019; Purushothaman et al., 2023; Wang & Wang, 2021) have predominantly focused on task-oriented settings, offering tailored solutions for individual distortions, whereas practical scenarios often involve multiple interacting degradations. Universal speech enhancement (USE) (Pascual et al., 2019; Nair & Koishida, 2021; Serrà et al., 2022; Scheibler et al., 2024) has emerged as a promising paradigm for addressing diverse distortions within a unified framework; however, reconciling task-specific precision with universal robustness remains a formidable challenge.

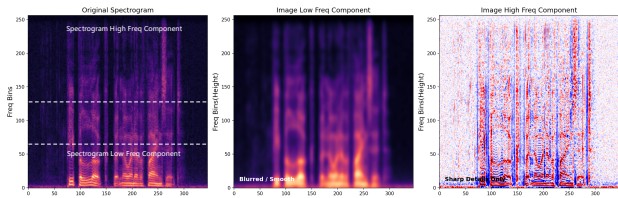

*Figure 1.* Dual-view of spectrogram frequency bands. Left: Acoustic perspective; Middle and Right: Visual image perspectives of low-frequency and high-frequency components, respectively.

Current SE methods are primarily bifurcated into two paradigms: predictive and generative. Predictive models (Krawczyk & Gerkmann, 2014; Williamson et al., 2015; Park et al., 2022; Lu et al., 2023; Abdulatif et al., 2024) typically formulate SE as a supervised learning task, optimizing either a direct mapping or a temporal-frequency mask to recover clean speech through deterministic objective functions. Although computational efficient and effective for single distortions, these methods often suffer from over-smoothing due to intrinsic "regression to the mean"

phenomenon, leading to the loss of fine-grained acoustic details, "muffled" outputs, and poor generalization in complex multi-distortion scenarios.

In contrast, generative methods model the intrinsic data distribution in a unified latent space, enabling effective recovery under severe information loss. Among various paradigms, including VAEs (Kingma & Welling, 2013; Richter et al., 2020), GANs (Goodfellow et al., 2014; Kong et al., 2020), and Normalizing Flows (Rezende & Mohamed, 2015; Yang et al., 2025a; Lee et al., 2025), Diffusion Models (Lu et al., 2022) have recently emerged as SOTA, driven by their success in image synthesis and their growing adoption in SE. Score-based diffusion models (Welker et al., 2022; Richter et al., 2023) are particularly notable for formulating diffusion via stochastic differential equations, enabling a continuous-time modeling framework. However, "pure" diffusion models entail prohibitive computational costs due to numerous iterative reverse steps. To mitigate this, recent work has shifted toward a predictive-diffusion paradigm (Lemercier et al., 2023; Scheibler et al., 2024; Zhang et al., 2025), which employs predictive models as robust priors to constrain the diffusion process, thereby enhancing reconstruction stability while reducing sampling overhead.

Despite these advancements, current methods still remain computationally burdensome. We attribute this inefficiency to a fundamental oversight: spectrograms are treated as generic 2D images with spatially uniform operations, ignoring their underlying physical structure. As illustrated in Fig. 1, we argue that extreme efficiency requires embracing the dual nature of spectrograms from two complementary perspectives: 1) a visual perspective, where spectrograms exhibit image-like textures and spatial patterns (Welker et al., 2022); 2) an acoustic perspective, where spectrograms are physical representations with strong anisotropy and non-uniform information density (Yu & Luo, 2023), featuring information-dense low-frequency harmonics and sparse yet critical high-frequency transients.

Based on above complementary views, we introduce DVPD, the dual-view predictive diffusion framework that harmonizes visual structural textures with acoustic physics. To optimize spectral efficiency, we design a frequency-adaptive non-uniform compression (FANC) encoder, which employs heterogeneous kernels to preserve low-frequency harmonic integrity while pruning high-frequency redundancies, in alignment with human auditory frequency resolution. Furthermore, we incorporate a backbone augmented with a multi-range lightweight image-based spectro-awareness (LISA) module to efficiently capture anisotropic features, including horizontal harmonic correlations and vertical transients, with minimal overhead. At inference time, we further propose a training-free lossless boost (TLB) strategy that exploits the dual-view structure to recalibrate feature scales,

yielding consistent quality improvements without additional training. Extensive experiments demonstrate that DVPD achieves a new efficiency and quality trade-off in generative SE. Notably, our model attains SOTA performance while using only 35% of the parameters and 40% of the MACs of the SOTA lightweight model, PGUSE, thereby establishing a new baseline for efficiency and quality in generative SE.

**Conflict of Interest Disclosure.** The authors declare no financial conflicts of interest related to this work.

## 2. Related Work

### 2.1. Score-based Diffusion Models for SE

Diffusion Probabilistic Models (DPMs) have revolutionized SE by mitigating the over-smoothing and loss of high-frequency details common in traditional methods (Lu et al., 2021; 2022). The field has matured with the introduction of continuous-time Stochastic Differential Equations (SDEs), enabling sophisticated score-based modeling in the complex STFT domain (Welker et al., 2022; Richter et al., 2023) and refined conditioning mechanisms (Tai et al., 2023; Yang et al., 2025b). Despite their generative prowess, these models incur prohibitive computational costs due to the extensive iterative sampling steps. While recent efforts like Latent Diffusion Models (LDM) (Zhao et al., 2025) attempt to mitigate this by operating in compressed spaces or optimizing trajectories, the inherent inference complexity remains a significant bottleneck.

### 2.2. Predictive-Diffusion Hybrid Architectures

To alleviate the sampling bottleneck and suppress hallucinations, recent trends have converged toward hybrid paradigms that integrate predictive and generative models. These approaches generally follow three trajectories: 1) Conditioning-based integration, where a predictive network serves as an auxiliary conditioner to guide the diffusion process (Serrà et al., 2022; Scheibler et al., 2024; Kim et al., 2024); 2) Cascaded regression refinement, where a predictive model provides an initial estimate or residual, which is then refined by the diffusion branch to reduce the required sampling steps (Qiu et al., 2023; Lemercier et al., 2023); 3) Parallel Architecture, which introduces a dual-branch parallel framework that achieves SOTA performance with relatively low complexity (Zhang et al., 2025). DVPD belongs to the parallel category but distinguishes itself through a domain-specific architecture. Instead of relying on generic spatially-homogeneous backbones, we explicitly model "spectrogram-image synergy" by bridging acoustic physics with visual structural priors, which allows DVPD to achieve superior performance with significantly fewer parameters and MACs.

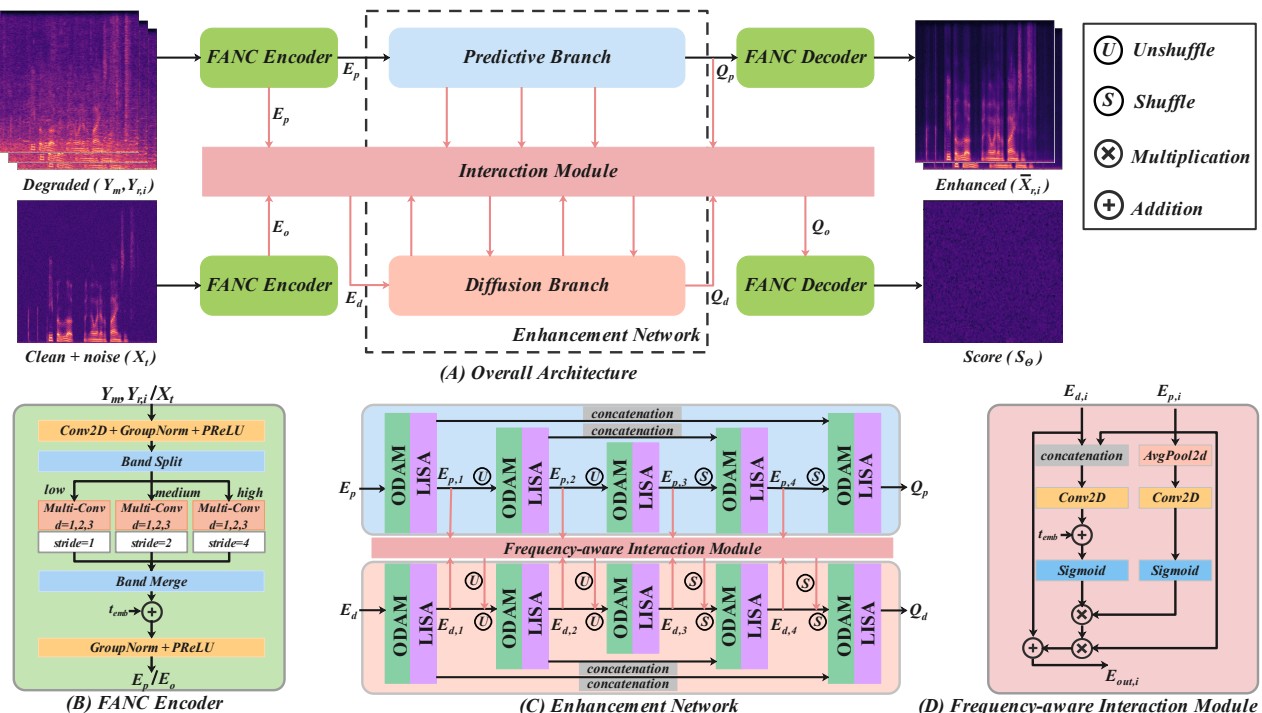

*Figure 2.* Architectural overview of the proposed DVPD. (A) The overall symmetrical dual-branch structure of DVPD, comprising the predictive branch (upper) and the diffusion branch (lower). (B) Detailed design of the Frequency-Adaptive Non-uniform Compression (FANC) Encoder. (C) Detailed design of enhancement network. (D) Illustration of the Frequency-Aware Interaction module.

## 3. The Proposed Model

### 3.1. Preliminaries: Score-based Diffusion Models

Score-based diffusion models conceptualize SE as a continuous time transformation between clean speech $\mathbf{x}_0 \in \mathbb{R}^{1 \times L}$ and a noisy condition $\mathbf{y} \in \mathbb{R}^{1 \times L}$ via Stochastic Differential Equations (SDEs) (Song et al., 2021). The forward process $\{\mathbf{x}_t\}_{t=0}^{T}$ perturbs the data distribution toward a prior distribution, while the reverse process recovers clean signals by learning the score function $\nabla_{\mathbf{x}_t} \log p_t(\mathbf{x}_t)$. In the context of SE, two prominent SDE formulations are widely used: *Ornstein-Uhlenbeck Variance Exploding* (OUVE) (Welker et al., 2022; Richter et al., 2023) and *Brownian Bridge with Exponential Diffusion* (BBED) (Lay et al., 2023). To mitigate the *prior mismatch* and inference bias inherent in finite-time OUVE formulations, we adopt the BBED. Unlike OUVE, where the mean only converges to $\mathbf{y}$ as $T \to \infty$, BBED's linear mean evolution ensures the forward process reaches $\mathbf{y}$ at $T = 1$, significantly enhancing restoration stability. A comprehensive mathematical comparison is provided in **Appendix A**.

### 3.2. Overall Pipeline

DVPD follows the parallel predictive-diffusion architecture illustrated in Fig. 2(A). All operations are conducted

in the T-F domain using STFT. Given a clean waveform $\mathbf{x}_0 \in \mathbb{R}^{1 \times L}$ and a degraded waveform $\mathbf{y} \in \mathbb{R}^{1 \times L}$, we obtain their complex spectrograms $\mathbf{X}_{r,i}, \mathbf{Y}_{r,i} \in \mathbb{R}^{2 \times F \times T}$, and the magnitude spectrum of them $\mathbf{Y}_m, \mathbf{X}_m \in \mathbb{R}^{1 \times F \times T}$. Power-law compression is then applied to compensate for the heavy-tailed distribution of speech amplitudes (Gerkmann & Martin, 2010) and improve numerical stability.

The predictive branch takes the degraded spectrogram $\mathbf{Y} = [\mathbf{Y}_{r,i}, \mathbf{Y}_m] \in \mathbb{R}^{3 \times F \times T}$ as input, which is initially projected by the *FANC encoder* into a compressed latent space to yield encoded features $\mathbf{E}_p \in \mathbb{R}^{C \times F_1 \times T}$. Subsequently, $\mathbf{E}_p$ is processed by the hierarchical *enhancement network* to produce $\mathbf{Q}_p \in \mathbb{R}^{C \times F_1 \times T}$, which is then passed through the *FANC decoder* to reconstruct the deterministic spectral estimate $\hat{\mathbf{X}}_{r,i} \in \mathbb{R}^{2 \times F \times T}$. Simultaneously, in the diffusion branch, the noise-perturbed magnitude $\mathbf{X}_t \in \mathbb{R}^{1 \times F \times T}$ is encoded into $\mathbf{E}_o \in \mathbb{R}^{C \times F_1 \times T}$ via the same FANC encoder. To leverage predictive guidance, an initial interaction module harmonizes $\mathbf{E}_o$ with $\mathbf{E}_p$ to produce a synergistic representation $\mathbf{E}_d \in \mathbb{R}^{C \times F \times T}$. This latent is then refined by the hierarchical enhancement network, where cross-branch interaction is performed at every level to inject stable deterministic priors into the diffusion flow. The resulting diffusion features $\mathbf{Q}_d$ are further calibrated with $\mathbf{Q}_p$ to yield $\mathbf{Q}_o \in \mathbb{R}^{C \times F_1 \times T}$, which is ultimately projected through the FANC decoder to estimate the score function $\mathbf{S}_\theta \in \mathbb{R}^{1 \times F \times T}$.

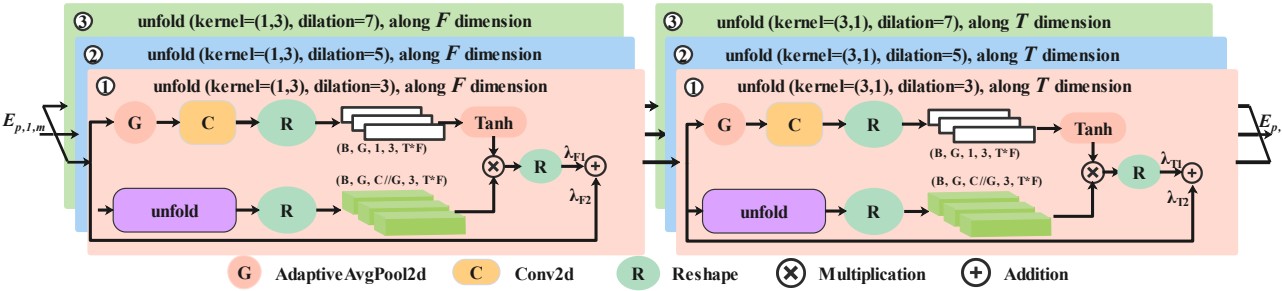

*Figure 3.* Detailed architecture of the LISA module.

Upon the estimation of the score function $\mathbf{S}_\theta$, the enhancement process concludes with a synergistic fusion and reconstruction phase. To ensure computational tractability, we execute a partial reverse diffusion process starting from a reduced time step $T_{\text{rs}}$, initialized as $\mathbf{X}_{T_{\text{rs}}} \approx \mu(\bar{\mathbf{X}}_m^p, \mathbf{Y}_m, T_{\text{rs}}) + \sigma(T_{\text{rs}})\mathbf{z}$. Throughout this iterative refinement, the TLB strategy (Sec. 3.4) is seamlessly integrated into the hierarchical U-Net backbone to adaptively modulate the feature maps. After $N$ sampling steps, the resulting generative magnitude $\hat{\mathbf{X}}_m^d$ is fused with the predictive magnitude $\bar{\mathbf{X}}_m^p$ to yield the final magnitude spectrum:

$$\hat{\mathbf{X}}_m = \alpha\bar{\mathbf{X}}_m^p + (1-\alpha)\hat{\mathbf{X}}_m^d, \qquad (1)$$

where $\alpha$ serves to balance deterministic stability with generative realism. The final complex spectrogram is then reconstructed by coupling $\hat{\mathbf{X}}_m$ with the phase spectrum inherited directly from the predictive branch, ultimately facilitating high-fidelity speech restoration with minimal computational overhead. A comprehensive summary of DVPD inference procedure is detailed in **Algorithm 1** (see Appendix).

### 3.3. Detailed Architectural Components

The core architecture of DVPD contains: FANC encoder, enhancement network, Frequency-aware Interaction (FI) Module, FANC decoder, which is described as follows:

**FANC Encoder** The FANC encoder (Fig. 2(B)) is designed to exploit the non-uniform information density of speech spectrograms, which distinguishes them from spatially homogeneous natural images. For inputs $\mathbf{Y} \in \mathbb{R}^{3 \times F \times T}$ or $\mathbf{X}_m \in \mathbb{R}^{1 \times F \times T}$, FANC implements a band-specific partitioning strategy: (*i*) low-band (0–2 kHz), containing the fundamental frequency ($f_0$) and primary formants, is preserved without compression to maintain critical harmonic integrity; (*ii*) mid-band (2–4 kHz) undergoes moderate compression; and (*iii*) high-band ($> 4$ kHz) is heavily compressed to prune spectral redundancy. In contrast to architectures like BSRNN (Yu & Luo, 2023) or PGUSE (Zhang et al., 2025) that often over-compress the low-frequency manifold, FANC prioritizes the preservation

of the "acoustic image." This is achieved through heterogeneous dilated kernels ($3 \times 3, 3 \times 5$, and $3 \times 7$) that create an *anisotropic receptive field* specifically targeting vertical transients and horizontal harmonics. For the predictive branch, this process yields the encoded features $\mathbf{E}_p \in \mathbb{R}^{C \times F_1 \times T}$. For the diffusion branch, sinusoidal time embeddings (Song et al., 2021) are integrated post-fusion, resulting in the generative encoded features $\mathbf{E}_o \in \mathbb{R}^{C \times F_1 \times T}$.

**Enhancement Network** To extract hierarchical acoustic features, we employ a 3-layer symmetric U-Net (Fig. 2(C)) as the core enhancement network. Each layer of the backbone is conceptualized as a global-to-local refinement unit, integrating the Omni-Directional Attention Mechanism (ODAM) (Ke et al., 2026) and the LISA module. Specifically, the predictive features $\mathbf{E}_p$ and diffusion features $\mathbf{E}_d$ are processed independently within each level before undergoing cross-branch calibration via the FI module. We utilize *Pixel Unshuffle* for downsampling and *Pixel Shuffle* for upsampling, effectively mitigating the information loss inherent in traditional pooling. At each level, downsampled features are fused with their counterparts through additive residuals to serve as input for the subsequent stage, ultimately yielding the refined hierarchical representations $\mathbf{Q}_p$ and $\mathbf{Q}_d$.

Taking the first predictive level as an exemplar, the feature $\mathbf{E}_p$ is first processed by the ODAM to capture global dependencies and model long-range correlations across both time and frequency axes. The resulting globally-aware feature $\mathbf{E}_{p,1,m}$ is then fed into the LISA module to refine spectral textures through a three-stage dynamic filtering process: (*i*) *Dynamic Kernel Generation*: instance-specific weights $\mathbf{W}_d = \tanh(\text{Conv}_{1\times1}(\text{GAP}(\mathbf{E}_{p,1,m})))$ are derived from the global context $\mathbf{E}_{p,1,m}$; (*ii*) *Stripe Dynamic Convolution*: anisotropic features are aggregated via $\mathbf{L}(\cdot) = \sum_{k=1}^{K} \mathbf{W}_d^{(k)} \odot \mathcal{U}(\text{Pad}(\cdot), d)^{(k)}$, where $\mathcal{U}(\cdot, d)$ denotes an unfold operation; and (*iii*) *Dual-path Refinement*: features are fused to balance structural stability and detail:

$$\begin{aligned} \mathbf{O}_{F,d} &= \lambda_{F1,d} \odot \mathbf{L}(\mathbf{E_{p,1,m}}) + \lambda_{F2,d} \odot \mathbf{E}_{p,1,m}, \\ \mathbf{N}_d &= \lambda_{T1,d} \odot \mathbf{L}(\mathbf{O_{F,d}}) + \lambda_{T2,d} \odot \mathbf{O}_{F,d}. \end{aligned} \qquad (2)$$

The final refined output $\mathbf{E}_{p,1}$ is formulated as $\mathbf{E}_{p,1} = \text{Conv}(\sum_{d \in \{3,5,7\}}(\gamma_d \Psi_d(\mathbf{E}_{p,1,m}) + \beta_d \mathbf{N}_d))$, where $\Psi_d(\cdot)$ represents sequential $\mathcal{T}$ and $\mathcal{F}$ stripe operations with multi-scale dilations $d \in \{3,5,7\}$.

**Frequency-aware Interaction (FI) Module**   The FI module (Fig. 2(D)) adaptively calibrates the cross-branch feature exchange through a dual-path gating mechanism. Given the diffusion feature $\mathbf{E}_{d,i}$, the predictive feature $\mathbf{E}_{p,i}$, and the time embedding $t_{emb}$, the interaction unfolds as follows: (*i*) *Joint Feature Fusion*: $\mathbf{E}_{d,i}$ and $\mathbf{E}_{p,i}$ are concatenated and processed via a *Conv2D* layer, with $t_{emb}$ added to inject temporal-diffusion context; (*ii*) *Global Prior Extraction*: a parallel path applies *AvgPool2d* and *Conv2D* to $\mathbf{E}_{p,i}$ to capture global spectral-temporal dependencies. Both paths are then passed through *Sigmoid* functions to generate dynamic importance masks, which are element-wise multiplied to produce a unified reliability weight. This weight modulates the predictive guidance $\mathbf{E}_{p,i}$ before it is residually added to the original diffusion latent. This process yields the final interaction output $\mathbf{E}_{out,i}$, ensuring that the generative process prioritizes robust low-frequency deterministic priors while effectively suppressing unreliable, noise-dominated regions in the predictive cues.

**FANC Decoder**   The FANC decoder symmetrically mirrors the partitioning structure of the encoder to facilitate resolution recovery. Taking the hierarchical features $\mathbf{Q}_p$ and $\mathbf{Q}_o$ as inputs, the decoder utilizes sub-pixel convolutions (SP-Conv2D) (Shi et al., 2016) to inversely map the compressed latent representations back to the original spectral resolution $F \times T$. This process ultimately yields the deterministic complex estimate $\hat{\mathbf{X}}_{r,i} \in \mathbb{R}^{2 \times F \times T}$ from the predictive branch and the score estimate $\mathbf{S}_\theta \in \mathbb{R}^{1 \times F \times T}$ from the diffusion branch.

### 3.4. TLB strategy

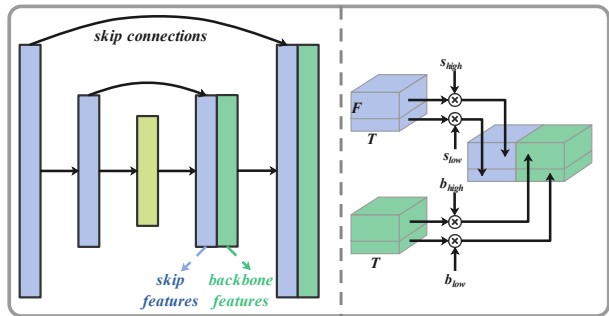

*Figure 4.* Operations of the TLB strategy. Scaling factors $b$ and $s$ modulate the intensity of backbone and skip features, respectively.

**TLB strategy**   Treating the spectrogram as a structural 2D manifold allows us to analyze the denoising dynamics

within the U-Net backbone. Recent studies (Si et al., 2024) suggest that the U-Net backbone inherently prioritizes low-frequency structural restoration while aggressively suppressing high-frequency noise. Conversely, skip connections are instrumental in recovering fine-grained transients but may inadvertently propagate residual noise or expedite premature convergence during the generative process. Building on our *dual-view* perspective, we propose the TLB strategy, an inference technique that modulates feature maps without additional training (Fig. 4). We define four scaling factors: $b_{low}, b_{high}$ (backbone) and $s_{low}, s_{high}$ (skip). The $b$ factors are formulated to modulate the backbone's high-frequency suppression intensity, whereas the $s$ factors recalibrate the spectral energy integrity of low-frequency harmonics within the skip connections. By aligning these factors with the spectrogram's anisotropic properties, where "low" and "high" (Fig. 1(left)) correspond to the distinct harmonic and transient regions of the spectrogram, TLB adaptively balances noise reduction and fine-grained detail preservation during the generative refinement stage. Please note that we only use the TLB strategy on the diffusion branch. For a more detailed description of using the spectrum as different frequency components for image denoising, see **Appendix B**.

## 4. Experiments

### 4.1. Datasets

To evaluate the efficacy and versatility of our model across diverse acoustic conditions, we conduct extensive experiments on several widely recognized benchmarks: We utilize the WSJ0-UNI (Ristea et al., 2025) dataset, which encompasses a broad range of distortions, to benchmark the model's performance on Universal Speech Enhancement (USE). The VoiceBank+DEMAND (VBDMD) (Botinhao et al., 2016) dataset is employed as a standard benchmark for evaluating localized noise suppression capabilities. VBDMD-SR for evaluating speech super-resolution. To further assess cross-task robustness, we evaluate our model on VBDMD-REVERB(VBD-RB), WSJ0-REVERB(WSJ0-RB) (Garofolo et al., 2007) and WSJ0-CHiME3(WSJ0-CE3) (Barker et al., 2015) to verify the model's out-of-distribution generalization in real world noisy and reverberant environments. All audio samples are resampled to 16 kHz for consistency. See **Appendix C** for more details.

### 4.2. Model Configurations and Evaluation Metrics

All audio signals were processed in the T-F domain using an STFT with a 512-point Hann window and a 128-point hop size. The U-Net backbone employed encoder/decoder stages with an initial channel 24, doubling at each level to 96 bottleneck. Each stage consisted of a single module, except for the diffusion bottleneck, which comprised two. For the BBED SDE, we set $k = 2.6$, $c = 0.51$, $T = 0.999$ (Lay

*Table 1.* Speech enhancement results where all models are both trained and evaluated on the WSJ0-UNI. **Bolds** indicate the best while underlines indicate the second best. MACs represent the total computational complexity of the entire inference process per second of audio.

| Method | Para. | MACs | Type | PESQ ↑ | ESTOI ↑ | CSIG ↑ | CBAK ↑ | COVL ↑ | WV-MOS ↑ |
|---|---|---|---|---|---|---|---|---|---|
| Degraded | - | - | - | 1.67 ± 0.60 | 0.70 ± 0.18 | 2.41 ± 1.15 | 1.92 ± 0.60 | 2.01 ± 0.87 | 1.79 ± 2.13 |
| Conv-TasNet (Luo & Mesgarani, 2019) | 3.4M | 3.2G | P | 2.01 ± 1.21 | 0.76 ± 0.09 | 3.02 ± 0.99 | 2.23 ± 0.78 | 2.64 ± 1.01 | 2.56 ± 1.33 |
| MANNER (Park et al., 2022) | 24.1M | 8.7G | P | 2.21 ± 1.23 | 0.80 ± 0.05 | 3.41 ± 0.66 | 2.46 ± 0.91 | 2.78 ± 1.03 | 2.99 ± 0.55 |
| PGUSE-P (Zhang et al., 2025) | 2.3M | 5.8G | P | 2.38 ± 1.10 | 0.85 ± 0.11 | 3.43 ± 0.73 | 2.60 ± 0.51 | 2.91 ± 0.61 | 3.21 ± 0.93 |
| CMGAN (Abdulatif et al., 2024) | 1.8M | 31.7G | P | 2.66 ± 0.99 | **0.88 ± 0.09** | 3.52 ± 0.81 | 2.81 ± 0.85 | 3.11 ± 0.62 | 3.55 ± 0.55 |
| MP-SENet (Lu et al., 2023) | 2.26M | 34.58G | P | **2.71 ± 0.89** | **0.88 ± 0.13** | **3.99 ± 0.76** | 2.90 ± 0.58 | **3.38 ± 0.89** | **4.16 ± 0.25** |
| DVPD-P (*ours*) | **0.61M** | **2.41G** | P | 2.70 ± 0.99 | 0.88 ± 0.08 | 3.91 ± 0.88 | 2.91 ± 0.59 | 3.28 ± 0.99 | 3.76 ± 0.47 |
| CDiffuSE (Lu et al., 2022) | 4.3M | 292.4G | D | 1.97 ± 0.91 | 0.80 ± 0.11 | 2.77 ± 0.71 | 1.99 ± 0.92 | 2.21 ± 1.01 | 2.50 ± 1.05 |
| SGMSE+ (Richter et al., 2023) | 65.6M | 8.0T | D | 2.61 ± 1.12 | 0.90 ± 0.11 | 3.79 ± 0.85 | 2.65 ± 0.81 | 3.09 ± 1.15 | 3.40 ± 0.91 |
| DOSE+ (Yang et al., 2025b) | 65.9M | 310.4G | D | 2.84 ± 0.66 | 0.90 ± 0.10 | 3.96 ± 0.63 | 2.79 ± 0.41 | 3.49 ± 0.79 | 3.69 ± 0.71 |
| StoRM (Lemercier et al., 2023) | 55.1M | 15.8T | D+P | 2.75 ± 1.03 | 0.89 ± 0.08 | 3.84 ± 0.77 | 2.66 ± 0.41 | 3.09 ± 0.90 | 3.44 ± 0.78 |
| UNIVERSE++ (Scheibler et al., 2024) | 42.9M | 42.8G | D+P | 2.66 ± 0.93 | 0.89 ± 0.11 | 3.89 ± 0.59 | 2.60 ± 0.49 | 3.21 ± 0.88 | 3.46 ± 0.77 |
| PGUSE (Zhang et al., 2025) | 5.1M | 26.3G | D+P | 2.95 ± 0.91 | 0.91 ± 0.06 | 4.01 ± 0.77 | 2.61 ± 0.60 | **3.53 ± 0.91** | 3.44 ± 0.66 |
| DVPD (*ours*) (*without TLB*) | **1.9M** | **10.2G** | D+P | 2.99 ± 0.88 | 0.91 ± 0.12 | 4.06 ± 0.71 | 2.93 ± 0.57 | 3.43 ± 0.87 | **4.16 ± 0.25** |
| DVPD (*ours*) (*with TLB*) | **1.9M** | **10.2G** | D+P | **3.15 ± 0.79** | **0.92 ± 0.05** | **4.21 ± 0.37** | **3.01 ± 0.47** | 3.51 ± 0.99 | **4.27 ± 0.31** |

et al., 2023), $\alpha = 0.4$, $T_{rs} = 0.12$ and $N = 3$ (**Appendix E**). The model was trained for 200 epochs using the AdamW on 4 NVIDIA RTX A6000 GPUs with a batch size of 32. The learning rate was $1 \times 10^{-3}$, decaying by 0.97 every two epochs, with gradient clipping at an $L_2$ norm of 5.0. The total loss $\mathcal{L}$ was a combination of predictive and generative objectives:

$$\mathcal{L} = \lambda_1 \mathcal{L}_{\text{mag}} + (1 - \lambda_1)\mathcal{L}_{\text{comp}} + \lambda_2 \mathcal{L}_{\text{pha}} + \mathcal{L}_{\text{score}}, \quad (3)$$

where $\mathcal{L}_{\text{mag}}$, $\mathcal{L}_{\text{comp}}$, and $\mathcal{L}_{\text{pha}}$ denoted the magnitude, complex spectral, and anti-wrapping phase losses for the predictive branch, respectively, and $\mathcal{L}_{\text{score}}$ was the score-matching loss. We empirically set $\lambda_1 = 0.5$ and $\lambda_2 = 0.002$ with many attempts to ensure that all loss components remained within the same order of magnitude during training.

Performance was evaluated using a comprehensive suite of metrics, including PESQ (Rix et al., 2001), ESTOI (Jensen & Taal, 2016), SI-SDR (Jonathan et al., 2019), WV-MOS (Andreev et al., 2023), DNS-MOS (Reddy et al., 2021) and the composite Mean Opinion Score (MOS) estimates (CSIG, CBAK, COVL) (Hu & Loizou, 2007). To demonstrate efficiency, we also reported the number of Parameters (M) and MACs [1]. More Details were provided in **Appendix D**.

### 4.3. Comparisons with Methods on USE

As demonstrated in Table 1, we evaluated the USE performance of our proposed model on the WSJ0-UNI dataset, comparing against various SOTA predictive and generative baselines in terms of parameter, computational complexity (MACs), and speech quality metrics. By decoupling our framework into its predictive and generative components,

we observe that the predictive branch, DVPD-P, achieves speech quality comparable to the leading MP-SENet while utilizing only one-third of its parameters and reducing MACs by an order of magnitude. In the diffusion domain, the results confirm that hybrid predictive-diffusion (D+P) frameworks consistently outperform pure diffusion (D) models; notably, compared to the previous SOTA lightweight model PGUSE, our DVPD requires only 35% of the parameters and 40% of the inference MACs while delivering superior performance across most objective measures. Furthermore, by incorporating our TLB strategy during inference, the performance of DVPD is significantly enhanced without any additional training or computational overhead, further validating the efficiency and robustness of our dual-view architectural design.

### 4.4. Evaluation of Out-of-Distribution Generalization

To evaluate robustness against distributional shifts, we assessed the zero-shot generalization of models trained exclusively on WSJ0-UNI (Fig. 5) across unseen speech sources, noise types, and reverberant profiles (VBDMD, VBD-RB, WSJ0-CE3, and WSJ0-RB). Several key observations emerge. First, diffusion-based models (solid lines) consistently exhibit superior generalization across nearly all benchmarks compared to predictive models (dashed lines), highlighting the inherent advantage of stochastic refinement in handling unseen distortions. Within the predictive category, our DVPD-P achieves performance comparable to the SOTA MP-SENet, even surpassing it in specific scenarios. This is particularly noteworthy given that DVPD-P possesses significantly fewer parameters and lower computational complexity. For the generative paradigm, our full DVPD model sets a new SOTA benchmark across the majority of datasets, further validating the efficacy of our

---

[1] https://github.com/sovrasov/flops-counter.pytorch

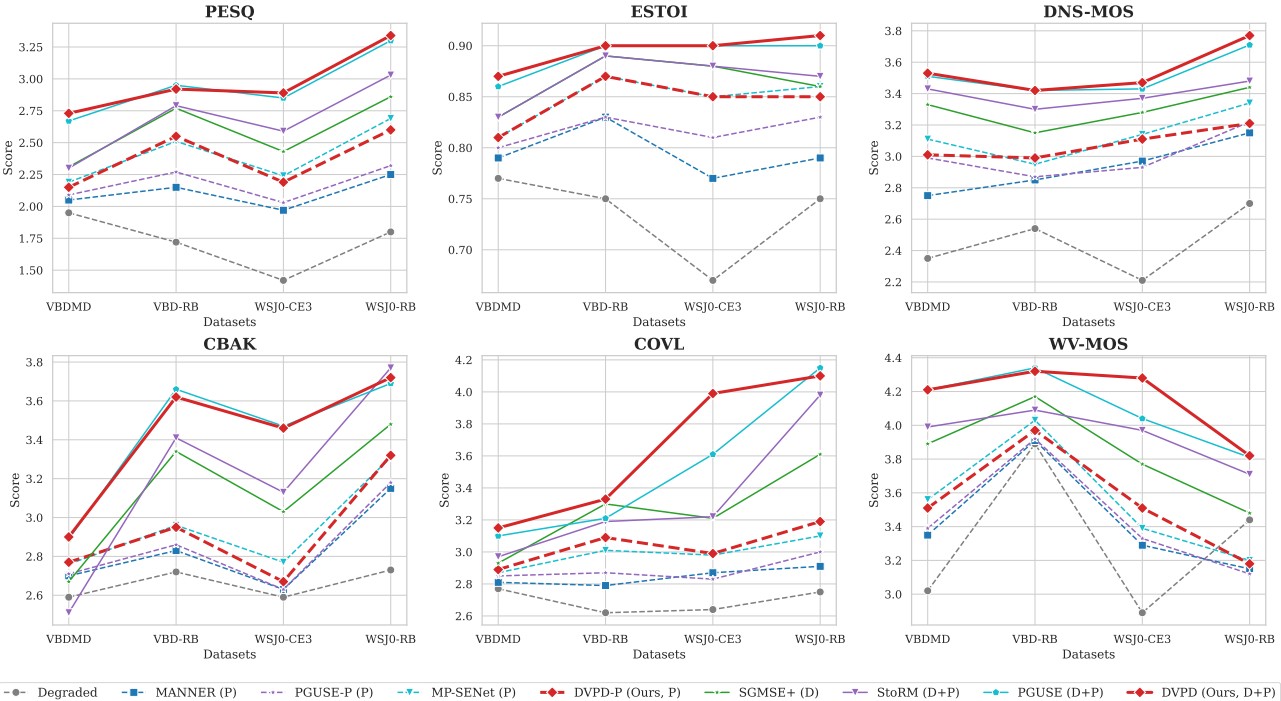

*Figure 5.* Performance comparison across diverse benchmarks. All models were trained exclusively on the WSJ0-UNI to evaluate their universal generalization capabilities. Dashed lines represent predictive models, while solid lines denote diffusion-based generative models.

dual-view architectural design in bridging the gap between efficiency and universal robustness.

### 4.5. Speech Denoising on VBDMD

Table 2 presents the performance of our model trained and evaluated specifically on the VBDMD benchmark. In this single distortion (noise-only) scenario, the spectral mapping task is relatively less complex compared to universal speech enhancement. Consequently, we observe that predictive models generally exhibit a performance edge over generative ones. Specifically, MP-SENet achieves the highest scores; this is likely because, in the absence of catastrophic information loss, the mean-regression characteristic of predictive models is highly effective at capturing the deterministic mapping between noisy and clean spectral manifolds.

Within this context, our predictive branch, DVPD-P, delivers performance second only to MP-SENet. More importantly, while pure generative models typically lag behind SOTA predictive models in localized denoising tasks, our DVPD (D+P) significantly narrows the performance gap between the two paradigms. By integrating the stable deterministic priors from the predictive branch, DVPD demonstrates exceptional versatility, proving its effectiveness even in simplified, single modality denoising scenarios.

*Table 2.* Performance comparison on the VBDMD dataset. All models are trained on the VBDMD. The symbol † denotes results obtained from our own implementations and evaluations.

| Method | PESQ ↑ | ESTOI ↑ | CSIG ↑ | DNS-MOS ↑ |
|---|---|---|---|---|
| Degraded | 1.98 | 0.79 | 3.48 | 2.39 |
| Conv-TasNet (Luo & Mesgarani, 2019) | 2.56 | 0.85 | 3.89 | 3.33 |
| PGUSE-P† (Zhang et al., 2025) | 3.09 | 0.87 | 4.45 | 3.59 |
| MP-SENet (Lu et al., 2023) | **3.50** | **0.91** | **4.73** | **3.67** |
| DVPD-P (*ours*) | 3.14 | 0.88 | 4.44 | 3.59 |
| CDiffuSE (Lu et al., 2022) | 2.48 | 0.79 | 3.77 | 3.31 |
| SGMSE+ (Richter et al., 2023) | 2.88 | 0.86 | 4.24 | 3.45 |
| StoRM (Lemercier et al., 2023) | 2.85 | 0.87 | 4.18 | 3.43 |
| UNIVERSE++ (Scheibler et al., 2024) | 3.03 | 0.87 | 4.38 | 3.51 |
| PGUSE† (Zhang et al., 2025) | 3.11 | 0.88 | 4.61 | 3.58 |
| FlowSE (Lee et al., 2025) | 3.12 | 0.88 | 4.62 | 3.58 |
| DVPD (*ours*) | **3.35** | **0.89** | **4.73** | **3.66** |

### 4.6. Evaluation on Speech Super-Resolution

We further evaluate the performance of DVPD on the VBDMD-SR dataset using the model trained on WSJ0-UNI. Speech super-resolution (SR), also known as bandwidth extension, is a widespread yet fundamentally ill-posed generative task that requires reconstructing high-frequency spectral content from band-limited observations. Compared to other baseline methods, DVPD maintains a leading position in most evaluation metrics, demonstrating its superior generative capacity. An interesting observation is that band-limited degraded speech achieves the highest PESQ score, while all enhanced models exhibit a performance decline in this

*Table 3.* Performance comparison on the VBDMD-SR dataset (8kHz - 16kHz). Models are trained on WSJ0-UNI.

| Method | PESQ ↑ | ESTOI ↑ | COVL ↑ | CSIG ↑ |
|---|---|---|---|---|
| Degraded | 4.22 | 0.95 | 2.99 | 1.69 |
| Conv-TasNet (Luo & Mesgarani, 2019) | 3.48 | 0.91 | 3.89 | 4.21 |
| MP-SENet (Lu et al., 2023) | **3.79** | 0.90 | 4.09 | **4.56** |
| DVPD-P (*ours*) | 3.70 | **0.91** | 4.15 | 4.39 |
| CDiffuSE (Lu et al., 2022) | 2.66 | 0.86 | 3.08 | 3.42 |
| SGMSE+ (Richter et al., 2023) | 3.84 | 0.92 | 3.91 | 3.86 |
| StoRM (Lemercier et al., 2023) | 2.89 | 0.88 | 3.24 | 3.50 |
| UNIVERSE++ (Scheibler et al., 2024) | 3.01 | 0.87 | 3.52 | 3.93 |
| PGUSE (Zhang et al., 2025) | 4.09 | 0.94 | 4.32 | 4.41 |
| DVPD (*ours*) | **4.15** | **0.96** | 4.28 | 4.44 |

*Table 4.* Ablation study on the WSJ0-UNI, evaluating the impact of key components and SDE formalisms. All results demonstrate the contribution of each module to the overall performance.

| Configuration | PESQ | CSIG | CBAK | WV-MOS |
|---|---|---|---|---|
| (DVPD) | **2.99** | **4.06** | **2.93** | **4.16** |
| w/o FANC Encoder | 2.93 | 3.98 | 2.86 | 4.09 |
| w/o FI Module | 2.91 | 4.01 | 2.88 | 4.12 |
| w/o Phase Loss | 2.91 | 3.99 | 2.88 | 4.11 |
| w/o LISA Module | 2.71 | 3.79 | 2.75 | 3.85 |
| w/ OUVE SDE | 2.89 | 3.95 | 2.81 | 4.06 |
| w/ Degraded Phase | 2.65 | 3.66 | 2.61 | 3.71 |

*Table 5.* Performance gain of the TLB strategy across different quality tiers on WSJ0-UNI and VBDMD benchmarks. Tiers are categorized by the baseline PESQ scores. All results are averaged over 20 independent inference runs to ensure statistical stability.

| Dataset | Performance Tier | Method | PESQ ↑ | ESTOI ↑ | SI-SDR | WV-MOS ↑ |
|---|---|---|---|---|---|---|
| WSJ0-UNI | (PESQ < 2) | Base | 1.73 | 0.75 | **16.99** | 3.12 |
| | | + TLB | **1.95** (*+0.22*) | **0.78** | 15.85 | **3.35** |
| | (2 ≤ PESQ < 3) | Base | 2.45 | 0.84 | **18.50** | 3.85 |
| | | + TLB | **2.51** (*+0.06*) | **0.85** | 18.42 | **3.92** |
| | (PESQ ≥ 3) | Base | 3.20 | 0.92 | **19.30** | 4.45 |
| | | + TLB | **3.24** (*+0.04*) | 0.92 | 19.25 | **4.46** |
| VBDMD | (PESQ < 2) | Base | 1.73 | 0.77 | **13.99** | 2.23 |
| | | + TLB | **1.88** (*+0.15*) | **0.78** | 12.43 | **2.41** |
| | (2 ≤ PESQ < 3) | Base | 2.59 | **0.82** | **17.29** | 3.35 |
| | | + TLB | **2.65** (*+0.06*) | **0.82** | 16.59 | **3.55** |
| | (PESQ ≥ 3) | Base | 3.56 | 0.92 | **20.81** | 4.58 |
| | | + TLB | **3.60** (*+0.04*) | 0.92 | 19.31 | **4.63** |

with the OUVE SDE. Furthermore, we observe that substituting our predicted phase with the original degraded phase for final waveform reconstruction leads to a substantial drop in speech quality. This validates our strategy of leveraging the predictive branch for high-fidelity phase estimation to guide the generative process.

specific metric. This phenomenon arises because the PESQ algorithm is not explicitly designed for SR evaluation, it is heavily biased toward low-frequency spectral fidelity, which dominates human auditory perception. During the generation process, models inevitably introduce minor reconstructive errors in the low-frequency regions, leading to a penalty in PESQ scores despite the successful reconstruction of high-frequency components.

### 4.7. Ablation Study

**Contribution of key components** To quantify the contribution of each key component within the DVPD framework, we conducted a comprehensive ablation study on the WSJ0-UNI dataset, as summarized in Table 4. Detailed configuration variants for each "without" (w/o) case are provided in **Appendix F**. From the architectural perspective, removing the FANC Encoder, the FI Module, or the Phase Loss leads to a measurable decrease in performance, although the impact is relatively moderate. However, the exclusion of the LISA Module results in a significant performance degradation across all perceptual metrics. This pronounced decline underscores the critical importance of our multi-range processing approach implemented via the LISA module. Regarding the diffusion formalism and phase strategy, replacing the BBED formulation with OUVE negatively impacts the efficacy of the model, which can be attributed to the inherent prior mismatch issue associated

**Analysis of the TLB Strategy** To further investigate the efficacy of the proposed TLB strategy, we stratify the test samples from WSJ0-UNI and VBDMD into three distinct quality tiers based on their baseline PESQ scores: (PESQ < 2), (2 ≤ PESQ < 3), and (PESQ ≥ 3). This stratification is motivated by the varying spectro-temporal characteristics across different quality levels. Accordingly, the scaling parameters $(s, b)$ are fine-tuned for each tier to achieve optimal restoration; The rationale for quality stratification and the sensitivity of parameters $(s, b)$ are further elaborated in **Appendix G**. As shown in Table 5, the TLB strategy yields the most substantial perceptual gains in the (PESQ < 2) across both datasets. This phenomenon can be attributed to the fact that in severely degraded scenarios, the base model often fails to fully reconstruct the fundamental harmonic manifold in the low-frequency regions. Our TLB strategy effectively compensates for these missing structural cues by recalibrating and injecting salient features from the skip-connections, thereby significantly enhancing the harmonic integrity and perceptual quality. Notably, the marginal SI-SDR decrease reflects a classic perception-fidelity tradeoff. While modulating skip-connection features enhances perceptual clarity, it inherently introduces minor sample-level variations, prioritizing spectral realism over strict waveform alignment.

**Transferability and Practical Tier Selection of TLB** The preceding analysis uses PESQ-based tiers to study where TLB is most effective. Since PESQ requires clean reference signals, this setting should be interpreted as an oracle analysis rather than a directly deployable selection rule.

*Table 6.* Transferability of the proposed TLB strategy to other diffusion-based baselines on WSJ0-UNI. DNSMOS is used as a practical reference-free tier selector.

| Model | Tier Selector | PESQ ↑ | DNSMOS ↑ |
|---|---|---|---|
| StoRM | None | 2.75 | 3.11 |
| StoRM + TLB | DNSMOS | 2.81 | 3.13 |
| PGUSE (U-Net) | None | 2.82 | 3.22 |
| PGUSE (U-Net) + TLB | DNSMOS | 3.02 | 3.55 |
| DVPD | None | 2.99 | 3.47 |
| DVPD + TLB | DNSMOS | **3.13** | **3.63** |

To examine whether TLB can be used in practical inference, we further replace PESQ with DNSMOS, a reference-free speech quality estimator, for tier selection.

Specifically, we empirically analyze the correspondence between DNSMOS and PESQ on WSJ0-UNI and VBDMD. We observe that samples with DNSMOS below 2.75 mostly fall into the low-PESQ regime, while samples with DNSMOS between 2.75 and 3.5 are mainly associated with medium-quality PESQ scores. Since the low-quality regime is also where TLB provides the largest gain, we adopt a slightly conservative DNSMOS-based deployment rule:

$$\begin{cases} \text{low-quality tier,} & \text{DNSMOS} < 2.5, \\ \text{medium-quality tier,} & 2.5 \leq \text{DNSMOS} < 3.5, \\ \text{high-quality tier,} & \text{DNSMOS} \geq 3.5. \end{cases}$$

This rule enables TLB to be applied without access to clean speech during inference.

We also evaluate whether TLB is specific to DVPD or can generalize to other U-Net-compatible diffusion backbones. The motivation of TLB is not tied to DVPD itself, but to a broader property of U-Net-based diffusion models: during reverse diffusion, low-frequency structures are usually restored more reliably, whereas high-frequency regions tend to be more conservative. Therefore, recalibrating early-stage skip features should be beneficial beyond a single backbone.

As shown in Table 6, applying TLB with DNSMOS-based tier selection consistently improves different diffusion baselines on WSJ0-UNI. For StoRM, which contains only a diffusion branch, TLB is applied to the first two U-Net stages, following the same setting as DVPD. For PGUSE, whose original implementation is not strictly U-Net-shaped, we construct a U-Net-compatible variant while keeping the remaining modules unchanged, and apply TLB to the first two stages of its diffusion branch. TLB improves StoRM from 2.75 to 2.81 PESQ and PGUSE (U-Net) from 2.82 to 3.02 PESQ. The gain on PGUSE (U-Net) is larger, possibly because its weaker baseline produces more low-quality samples, where TLB is more effective. In contrast, StoRM shows a smaller gain, which may be due to its deeper U-Net backbone reducing the relative influence of modulation

applied only at the first two stages.

Overall, these results support two conclusions. First, TLB is a transferable inference-time booster for U-Net-compatible diffusion models, especially when the baseline outputs contain more low-quality samples. Second, the advantage of DVPD is not solely due to TLB: even under DNSMOS-based practical tier selection, DVPD + TLB still achieves the best final performance among all compared systems.

## 5. Conclusion

We presented DVPD, a novel predictive-diffusion framework that leverages a dual-view perspective to model spectrograms as both physical frequency representations and visual textures. Through the FANC encoder, LISA module, and the TLB strategy, our model significantly reduces computational overhead while improving restoration quality. Experimental results show that DVPD outperforms much larger SOTA models using only 35% of the parameters and 40% of the MACs, which provides a robust solution for real-world SE.

## Acknowledgements

This paper is supported by Joint Funds of the National Natural Science Foundation of China (NSFC) No. U25A20390, National Key Research and Development Program of China, and Guangdong Basic and Applied Basic Research Foundation (2026A1515010215).

## Impact Statement

This work aims to improve the efficiency and robustness of speech enhancement systems, which may benefit real-world speech communication, automatic speech recognition, and assistive listening applications. More efficient models can reduce computational cost and make speech enhancement more accessible on resource-constrained devices. At the same time, improved speech processing systems may raise privacy concerns if deployed without user consent, especially in scenarios involving personal conversations or sensitive acoustic environments. We encourage responsible deployment with clear user consent, privacy-preserving data handling, and appropriate safeguards against misuse. The proposed method is intended for beneficial speech processing applications rather than surveillance or unauthorized speech monitoring.

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

# A. Extended Background on Score-based Diffusion Models

## A.1. Forward and Reverse SDEs

The forward diffusion process $\{\mathbf{x}_t\}_{t=0}^{T}$ is defined by the following Itô SDE:

$$d\mathbf{x}_t = \mathbf{f}(\mathbf{x}_t, t)dt + g(t)d\mathbf{w}, \tag{4}$$

where $\mathbf{f}(\cdot, t)$ is the drift coefficient, $g(t)$ is the diffusion coefficient, and $\mathbf{w}$ denotes the standard $d$-dimensional Brownian motion. In SE, this process is applied independently to each T-F bin in the STFT domain.

Under certain regularity conditions, the reverse-time process $\{\mathbf{x}_t\}_{t=T}^{0}$ satisfies (Anderson, 1982; Song et al., 2021):

$$d\mathbf{x}_t = [-\mathbf{f}(\mathbf{x}_t, t) + g^2(t)\nabla_{\mathbf{x}_t} \log p_t(\mathbf{x}_t)]dt + g(t)d\bar{\mathbf{w}}, \tag{5}$$

where $d\bar{\mathbf{w}}$ is the reverse-time Brownian motion. The term $\nabla_{\mathbf{x}_t} \log p_t(\mathbf{x}_t)$ is the score function, which we approximate using a neural network $\mathbf{s}_\theta(\mathbf{x}_t, \mathbf{y}, t)$ trained via denoising score matching:

$$\mathcal{L}_{\text{score}} = \mathbb{E}_{t,\mathbf{x}_0,\mathbf{y},\mathbf{z}} \left[ \|\mathbf{s}_\theta(\mathbf{x}_t, \mathbf{y}, t) + \frac{\mathbf{z}}{\sigma(t)}\|_2^2 \right], \tag{6}$$

where $\mathbf{z} \sim \mathcal{N}(0, \mathbf{I})$ and $\mathbf{x}_t = \mu(\mathbf{x}_0, \mathbf{y}, t) + \sigma(t)\mathbf{z}$.

## A.2. Comparison: OUVE vs. BBED

**OUVE formulation:** The drift and diffusion coefficients for OUVE are defined as:

$$\mathbf{f}(\mathbf{x}_t, t) = \gamma(\mathbf{y} - \mathbf{x}_t), \quad g(t) = \sqrt{ck^t}, \tag{7}$$

where $\gamma > 0$ is the stiffness parameter that controls the speed of the mean transition from $\mathbf{x}_0$ to $\mathbf{y}$. The parameters $c > 0$ and $k > 1$ are hyperparameters governing the noise schedule, where $c$ scales the diffusion intensity and $k$ determines the exponential growth rate of the noise variance over time. Under these definitions, the closed-form mean and variance are:

$$\mu(\mathbf{x}_0, \mathbf{y}, t) = e^{-\gamma t}\mathbf{x}_0 + (1 - e^{-\gamma t})\mathbf{y}, \tag{8}$$

$$\sigma^2(t) = \frac{c(k^t - e^{-2\gamma t})}{2\gamma + \log k}. \tag{9}$$

**BBED formulation:** BBED resolves the prior mismatch by redefining the drift coefficient:

$$\mathbf{f}(\mathbf{x}_t, t) = \frac{\mathbf{y} - \mathbf{x}_t}{1 - t}, \tag{10}$$

while maintaining the same diffusion coefficient $g(t)$ to ensure a consistent noise schedule for fair comparison. The linear mean evolution is given by:

$$\mu(\mathbf{x}_0, \mathbf{y}, t) = (1 - t)\mathbf{x}_0 + t\mathbf{y}. \tag{11}$$

As $t \to 1$, the mean exactly reaches $\mathbf{y}$, effectively eliminating the prior mismatch encountered in OUVE. To ensure numerical stability and avoid the singularity at $t = 1$, we set the terminal diffusion time $T = 0.999$.

## A.3. Sampling via Euler-Maruyama

During inference, we discretize the interval $[0, T]$ into $N$ sub-intervals with step size $\Delta t = T/N$. The reverse process is approximated using the Euler-Maruyama method:

$$\mathbf{x}_{t-\Delta t} = \mathbf{x}_t - [-\mathbf{f}(\mathbf{x}_t, t) + g^2(t)\mathbf{s}_\theta(\mathbf{x}_t, \mathbf{y}, t)]\Delta t + g(t)\sqrt{\Delta t}\mathbf{z}, \tag{12}$$

where $\mathbf{z} \sim \mathcal{N}(0, \mathbf{I})$. This approach allows us to generate clean samples starting from the initial state $\mathbf{x}_T \sim \mathcal{N}(\mathbf{y}, \sigma^2(T)\mathbf{I})$.

*Table 7.* Distortion categories and corresponding probabilities in the synthetic training set.

| Family | Distortion Type | Prob. |
|---|---|---|
| Noise | General additive noise | 0.50 |
| | Gaussian white noise | 0.60 |
| Reverb | Freeverb / RIR convolution | 0.20 |
| Microphone | Bandpass filtering | 0.20 |
| | Bad mic frequency response | 0.50 |
| ADC/DAC | High-pass filtering | 0.70 |
| | Low-pass filtering | 0.70 |
| | Bit depth reduction (8-24 bit) | 0.10 |
| AGC | Dynamic range expansion | 1.00 |
| | Post-processing gain | 0.10 |
| Preprocessing | Hard clipping (Hard limit) | 0.25 |
| | Post-clipping gain change | 0.25 |
| | Resampling (3-32 kHz) | 0.40 |
| | Multi-algorithm resampling | 1.00 |
| Transmission | GSM network compression | 0.25 |
| Misc. | Speaker gain fluctuation | 0.20 |
| | Nearend gain change | 0.20 |
| | Phaser (Phase distortion) | 0.02 |
| | Non-linear Tanh distortion | 0.01 |

## B. Design Philosophy of the TLB Strategy

### B.1. Motivation for Frequency-Band Partitioning

The core logic for splitting the scaling parameters $s$ and $b$ into "high" and "low" frequency components is rooted in the distinct denoising dynamics of spectrograms when treated as 2D images. As illustrated in Fig. 6, during the reverse diffusion process, high-frequency regions typically exhibit rapid denoising, whereas low-frequency structural components evolve more gradually. From an acoustic perspective, we define high-quality restoration via two dimensions:

- **Harmonic Integrity (The "Comb" Structure):** In the low-to-mid frequency regions, clean speech is characterized by sharp, horizontal "comb-like" stripes representing the fundamental frequency ($f_0$) and its harmonics. Clear energy valleys (low-energy gaps) between these stripes are indicative of superior pitch restoration.

- **Consonant Texture (High-Frequency Transients):** High-frequency regions (the upper portion of the Y axis) consist of vertical, stochastic yet bounded energy blocks representing consonants. Precise reconstruction requires sharp temporal boundaries and distinct textures rather than smeared background noise.

We empirically set $2\,\text{kHz}$ (Schroter et al., 2022; Yu & Luo, 2023) (corresponding to Frequency Bin 64 in our 512 points STFT) as the boundary. This threshold is chosen because the $0$–$2\,\text{kHz}$ range encompasses the $f_0$ and the majority of critical formants, accounting for over 80% of the total spectral energy and speech information. By splitting the parameters at this boundary, TLB can independently recalibrate the backbone's high-frequency suppression and the skip-connections' low-frequency structural compensation.

### B.2. Parameter Selection and Branch-Specific Application

Our 3-layer U-Net backbone consists of two encoder-decoder levels and a bottleneck. The TLB strategy is applied to the first two levels, resulting in a total of eight parameters:

- **Level 1:** $s_{1,\text{high}}, s_{1,\text{low}}$ (skip) and $b_{1,\text{high}}, b_{1,\text{low}}$ (backbone).

- **Level 2:** $s_{2,\text{high}}, s_{2,\text{low}}$ (skip) and $b_{2,\text{high}}, b_{2,\text{low}}$ (backbone).

Notably, the TLB strategy is exclusively applied to the diffusion branch. Although both the predictive and diffusion branches utilize U-Net architectures, the predictive branch processes complex-valued spectrograms (incorporating real, imaginary,

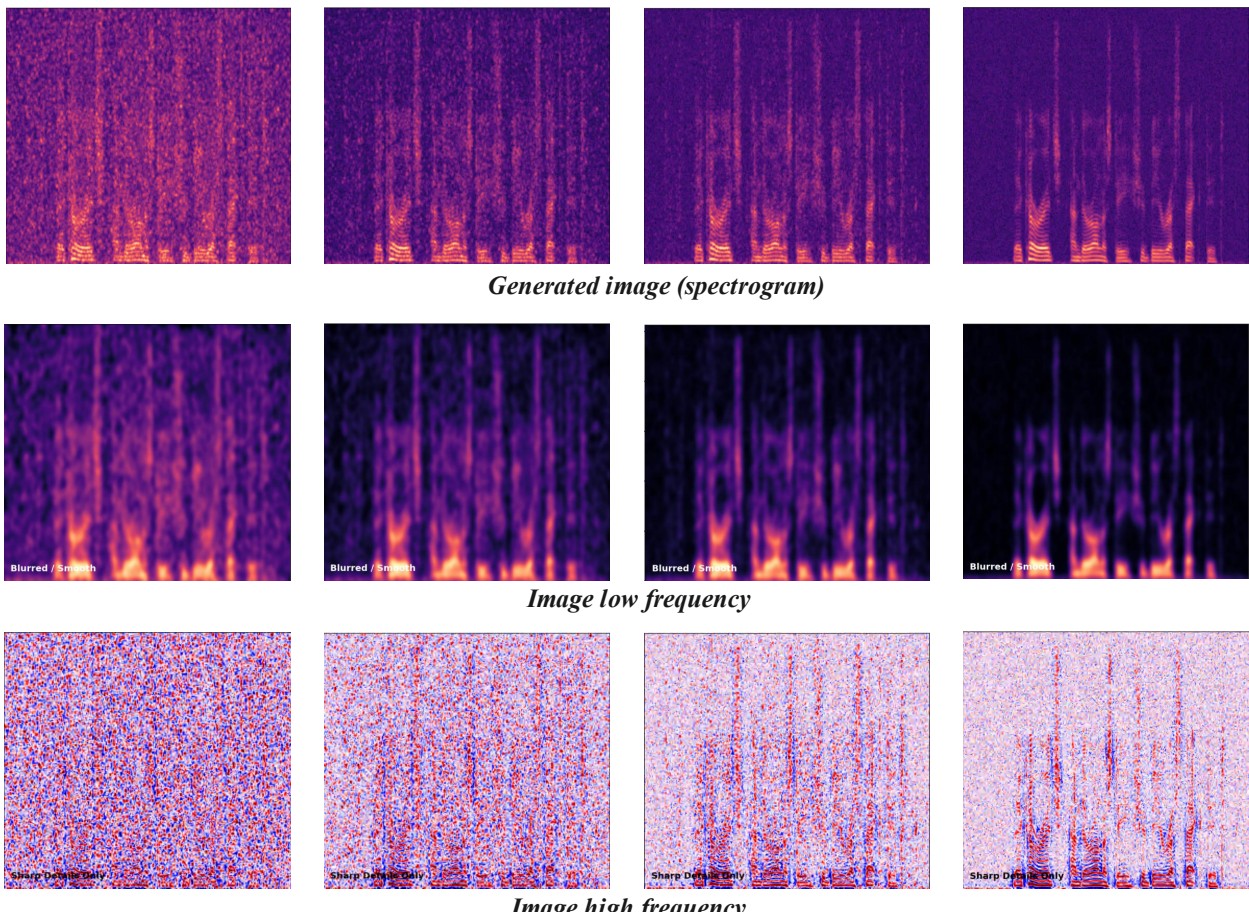

*Generated image (spectrogram)*

*Image low frequency*

*Image high frequency*

*Figure 6.* The reverse denoising process. The top row illustrates the image's progressive denoising process across iterations, while the subsequent two rows display low-frequency and high-frequency components.

and magnitude components). Direct numerical modulation of these feature maps during inference can inadvertently disrupt the delicate relationship between phase and magnitude, leading to reconstructive errors and performance degradation. In contrast, the diffusion branch operates on a score-based manifold that is better suited for such spectral-image recalibration, allowing TLB to enhance perceptual quality without compromising phase consistency. Specifically, a value of $s$ **exceeding unity** ($s > 1$) amplifies the injection of high-frequency details from the skip-connections, whereas a value **below unity** ($s < 1$) attenuates this contribution, reducing the fine-grained spectral textures. Similarly, $b$ regulates the backbone's denoising intensity: $b > 1$ enhances the noise suppression capability, while $b < 1$ mitigates the denoising effect to prevent over-smoothing. Therefore, the strategic adjustment of $s$ and $b$ is crucial for the TLB strategy to adaptively balance detail restoration and interference rejection across varying signal-to-noise ratios.

## C. Detailed Dataset Descriptions

In this section, we provide the implementation details and synthesis protocols for the datasets used in our evaluation.

**WSJ0-UNI:** Following the pipeline established by the Speech Signal Improvement Challenge (Ristea et al., 2025), we synthesized the WSJ0-UNI dataset to evaluate the model's capacity for universal restoration. We extended the diversity of distortions compared to previous works to include: recorded ambient noise, room reverberation, microphone frequency response variations, analog-to-digital converter (ADC) effects, automatic gain control (AGC) artifacts, and transmission-induced signal degradations (Table 7). Clean speech are sourced from the Wall Street Journal (WSJ0) corpus (Garofolo et al., 2007). We utilized the si_tr_s set for training, and si_dt_05 and si_et_05 for validation and testing, respectively.

---

**Algorithm 1** Inference of DVPD with TLB Strategy

---

1: **Input:** Degraded components $\{Y_r, Y_i, Y_m\}$; Starting time $T_{rs}$; Steps $N$; Fusion factor $\alpha$; TLB factors $\mathcal{S} = \{s_{l,band}\}$, $\mathcal{B} = \{b_{l,band}\}$.
2: **Output:** Enhanced components $\{\hat{X}_r, \hat{X}_i\}$.
3: *Step 1: Predictive Prior Generation*
4: $[\bar{X}_r^p, \bar{X}_i^p], \{h_l^p\}_{l=0}^5 \leftarrow P_\theta([Y_r, Y_i, Y_m])$
5: $\bar{X}_m^p \leftarrow \sqrt{(\bar{X}_r^p)^2 + (\bar{X}_i^p)^2}$
6: $\Phi^p \leftarrow \text{atan2}(\bar{X}_i^p, \bar{X}_r^p)$
7: *Step 2: Diffusion Initialization*
8: $X_{T_{rs}} \leftarrow \mu(\bar{X}_m^p, Y_m, T_{rs}) + \sigma(T_{rs})\mathbf{Z}, \quad \mathbf{Z} \sim \mathcal{N}(0, \mathbf{I})$
9: *Step 3: Generative Refinement with TLB*
10: **for** $t = T_{rs}, T_{rs} - \Delta t, \ldots, \Delta t$ **do**
11: $\quad$ *Inside $G_\theta(X_t, \{h_l^{pred}\}, t)$ with TLB recalibration:*
12: $\quad$ **for** Level $l \in \{1, 2\}$ **do**
13: $\quad\quad x_{skip,l} \leftarrow \text{Encoder}_l(X_t)$
14: $\quad\quad x_{skip,l}(\mathcal{F}_{low}) \leftarrow x_{skip,l}(\mathcal{F}_{low}) \cdot s_{l,low}$
15: $\quad\quad x_{skip,l}(\mathcal{F}_{high}) \leftarrow x_{skip,l}(\mathcal{F}_{high}) \cdot s_{l,high}$
16: $\quad$ **end for**
17: $\quad$ **for** Level $l \in \{1, 2\}$ **do**
18: $\quad\quad x_{back,l} \leftarrow \text{Decoder}_l(\cdot)$
19: $\quad\quad \mathbf{m}_l \leftarrow \text{Normalize}(\text{Mean}(x_{back,l}))$
20: $\quad\quad x_{back,l}(\mathcal{F}_{low}) \leftarrow x_{back,l}(\mathcal{F}_{low}) \odot ((b_{l,low} - 1) \cdot \mathbf{m}_l + 1)$
21: $\quad\quad x_{back,l}(\mathcal{F}_{high}) \leftarrow x_{back,l}(\mathcal{F}_{high}) \odot ((b_{l,high} - 1) \cdot \mathbf{m}_l + 1)$
22: $\quad$ **end for**
23: $\quad s_\theta \leftarrow \text{Score}(x_{back,1}, \ldots)$
24: $\quad X_{mean} \leftarrow X_t + [-f(X_t, t) + g(t)^2 s_\theta]\Delta t$
25: $\quad X_{t-\Delta t} \leftarrow X_{mean} + g(t)\sqrt{\Delta t}\mathbf{Z}$
26: **end for**
27: *Step 4: Synergistic Magnitude Fusion*
28: $\hat{X}_m^d \leftarrow \text{Clip}(X_{mean}, 0, +\infty)$
29: $\hat{X}_m \leftarrow \alpha\bar{X}_m^p + (1 - \alpha)\hat{X}_m^d$
30: *Step 5: Waveform Reconstruction*
31: $\hat{X}_r, \hat{X}_i \leftarrow \hat{X}_m \cos(\Phi^p), \hat{X}_m \sin(\Phi^p)$
32: **return** $\{\hat{X}_r, \hat{X}_i\}$

---

Noise clips were randomly sampled from the WHAM! dataset (Wichern et al., 2019).

**VoiceBank+DEMAND (VBDMD)** As a standard benchmark for monaural speech denoising, we utilized the publicly available VBDMD dataset (Botinhao et al., 2016). The training set includes 11,572 utterances from 28 speakers, while the test set consists of 872 utterances from 2 distinct speakers (non-overlapping with the training set). Clean signals were mixed with DEMAND (Thiemann et al., 2013) noise at Signal-to-Noise Ratios (SNRs) of $\{0, 5, 10, 15\}$ dB for training and $\{2.5, 7.5, 12.5, 17.5\}$ dB for testing.

**VBDMD-REVERB (VBD-RB)** To assess the model's generalization to unseen reverberant environments, we created the VBDMD-REVERB set by applying a stereo reverberation algorithm (Schroeder & Logan, 2003) to the clean test utterances of VBDMD. The resulting average reverberation time ($T_{60}$) is 0.4 s, providing a robust evaluate for dereverberation capabilities without specific training on this set.

**VBDMD-SR (Bandwidth Extension)** Speech Super-Resolution (SR), or bandwidth extension, requires the model to reconstruct high-frequency spectral content from band-limited signals. We simulated this by applying a 12th-order Butterworth low-pass filter with a cutoff frequency of 4 kHz to the VB-DMD test set. This task evaluates the generative strength of our dual-view framework in recovering high-frequency harmonic structures.

**WSJ0-CHiME3 (WSJ0-CE3)**    To rigorously assess the model's robustness against out-of-distribution (OOD) real-world noise, we synthesized the WSJ0-CE3 benchmark by combining clean utterances with authentic urban noise recordings. The clean components are sourced from the Wall Street Journal (WSJ0) corpus (Garofolo et al., 2007), a gold-standard benchmark in speech processing. It consists of high-fidelity, read-out news text recorded in controlled, anechoic-like environments using close-talk microphones. This ensures that the base signal is devoid of any pre-existing background noise or reverberation, providing a clear ground truth for reconstruction. The noise signals are extracted from the third CHiME challenge (CHiME-3) dataset (Barker et al., 2015). Unlike synthetic or stationary noise, CHiME-3 features multi-channel recordings captured in four highly dynamic and challenging urban environments: *public transport (Bus)*, *busy cafes (Cafe)*, *pedestrian areas (Pedestrian)*, and *street junctions (Street)*. These recordings contain highly non-stationary acoustic events, such as background chatter, engine rumbles, and sudden clatter, which are representative of the complex distortions encountered in daily communication. For our evaluation, we randomly selected clean utterances from the WSJ0 `si_et_05` set and mixed them with noise clips from the CHiME-3 test partition. To simulate varying levels of signal degradation, the Signal-to-Noise Ratio (SNR) was uniformly sampled from a wide range of $-6$ to $14$ dB. This setup, particularly at negative SNR levels, represents severe acoustic conditions that demand high generative capability to restore intelligibility. All mixed signals were resampled to $16$ kHz.

**WSJ0-REVERB (WSJ0-RB)** We simulated realistic reverberant scenarios using the *pyroomacoustics* engine. Room dimensions were uniformly sampled within $[5, 15] \times [5, 15] \times [2, 6]$ m, with $T_{60}$ values ranging from 0.4 to 1.0 s. This resulted in a mean Direct-to-Reverberant Ratio (DRR) of approximately $-9$ dB and a measured average $T_{60}$ of 0.91 s.

## D. Training Objectives and Evaluation Metrics

### D.1. Loss Functions for the Predictive Branch

To ensure the predictive branch provides high-quality phase and magnitude estimates, we employ a combination of spectral and phase-aware losses.

**Spectral Losses**    The magnitude and complex-domain Mean Square Errors (MSE) are defined as:

$$\mathcal{L}_{\text{mag}} = \mathbb{E}[\|\hat{X}_{\text{pred},m} - X_m\|_2^2], \tag{13}$$

$$\mathcal{L}_{\text{comp}} = \mathbb{E}[\|\hat{X}_{\text{pred},r} - X_r\|_2^2 + \|\hat{X}_{\text{pred},i} - X_i\|_2^2]. \tag{14}$$

**Anti-wrapping Phase Loss**    Given that the final enhanced waveform inherits the phase from the predictive branch, we introduce an anti-wrapping phase loss $\mathcal{L}_{\text{pha}}$ (Lay et al., 2023) to mitigate phase discontinuities:

$$\mathcal{L}_{\text{pha}} = \mathcal{L}_{\text{IP}} + \mathcal{L}_{\text{GD}} + \mathcal{L}_{\text{IAF}}, \tag{15}$$

where $\mathcal{L}_{\text{IP}}$, $\mathcal{L}_{\text{GD}}$, and $\mathcal{L}_{\text{IAF}}$ denote the instantaneous phase, group delay, and instantaneous angular frequency losses. To handle the periodicity of phase, we utilize the anti-wrapping function $f_{\text{AW}}(t) = |t - 2\pi \cdot \text{round}(t/2\pi)|$. These losses are computed using differential operators along the frequency and time axes to ensure spectro-temporal phase consistency.

### D.2. Detailed Definitions of Evaluation Metrics

To provide a multifaceted evaluation of the proposed DVPD, we employ a suite of objective metrics encompassing perceptual quality, intelligibility, signal fidelity, and deep learning based subjective score estimation.

**PESQ (Perceptual Evaluation of Speech Quality):**    Validated by ITU-T Recommendation P.862 (Rix et al., 2001), PESQ is the most widely used intrusive metric for assessing the quality of narrow-band and wide-band speech. It models the human auditory system by comparing the loudness spectra of the reference and degraded signals. In this paper, we report the Wideband PESQ (P.862.2) scores, ranging from 0.5 to 4.5, where higher scores indicate superior perceptual quality.

**ESTOI (Extended Short-Time Objective Intelligibility):**    ESTOI (Jensen & Taal, 2016) is an extension of the STOI metric designed to predict the intelligibility of speech in the presence of highly non-stationary noise. Unlike STOI, ESTOI accounts for the dependencies between frequency bands, making it more robust for evaluating complex restoration tasks. The score ranges from 0 to 1, representing the percentage of words likely understood by a human listener.

**SI-SDR (Scale-Invariant Signal-to-Distortion Ratio):** SI-SDR (Jonathan et al., 2019) is an energy-ratio-based metric that measures global reconstruction fidelity. Crucially, it is invariant to the gain difference between the estimated and reference signals, which is particularly important for generative models that may produce variations in signal amplitude. It is defined as:

$$\text{SI-SDR} = 10 \log_{10} \left( \frac{\|\mathbf{s}_{\text{target}}\|^2}{\|\mathbf{e}_{\text{noise}}\|^2} \right),$$
(16)

where $\mathbf{s}_{\text{target}}$ is the orthogonal projection of the estimated signal onto the clean reference.

**WV-MOS (Wav2vec MOS):** WV-MOS (Andreev et al., 2023) is a non-intrusive metric based on a pre-trained Wav2Vec 2.0 model. It is trained on large-scale subjective datasets to predict human Mean Opinion Scores directly from raw waveforms. It has shown a high correlation with subjective listening tests in recent generative speech enhancement literature.

**DNS-MOS (Deep Noise Supression MOS):** The DNS-MOS metric (Reddy et al., 2021) was developed by Microsoft based on the ITU-T P.808 standard. It utilizes a deep neural network to predict subjective quality for noise suppression tasks. DNS-MOS provides a robust estimation of perceptual quality in diverse real-world environments, making it a critical benchmark for the Universal Speech Enhancement (USE) scenarios investigated in this work.

**Composite Metrics (CSIG, CBAK, COVL):** Following the methodology in (Hu & Loizou, 2007), we report three composite Mean Opinion Score (MOS) estimates:

- **CSIG:** Predicts the MOS of signal distortion, focusing on the integrity of the speech itself.

- **CBAK:** Predicts the MOS of background noise intrusiveness, measuring the suppression of residual noise.

- **COVL:** Predicts the overall MOS, representing the comprehensive quality of the enhanced audio.

All composite metrics range from 1 to 5.

## E. Sensitivity Analysis of Inference Hyperparameters

In this section, we provide detailed ablation studies on the key inference hyperparameters: the weighting factor $\alpha$, the starting time step $T_{rs}$, and the number of reverse sampling steps $N$. All experiments in this section were conducted on the WSJ0-UNI test set.

### E.1. Balance between Predictive and Generative Branches ($\alpha$)

The parameter $\alpha$ weights the magnitude spectra from the predictive ($\hat{\mathbf{X}}_{\text{p},m}$) and diffusion ($\hat{\mathbf{X}}_{\text{d},m}$) branches. As shown in Table 8, when $\alpha = 1.0$, the model relies solely on the predictive branch, leading to higher SI-SDR but lower PESQ due to spectral over-smoothing. Conversely, a very small $\alpha$ (e.g., 0.2, 0) introduces more generative details but slightly degrades signal fidelity. We empirically set $\alpha = 0.4$ as it yields the optimal balance between perceptual quality (PESQ) and objective fidelity.

*Table 8.* Ablation of the weighting factor $\alpha$ ($T_{rs} = 0.12, N = 3$).

| $\alpha$ | PESQ ↑ | SI-SDR ↑ | ESTOI ↑ | WV-MOS ↑ | DNS-MOS ↑ |
|---|---|---|---|---|---|
| 1.0 (Pred. Only) | 2.70 | **20.55** | 0.88 | 3.76 | 3.42 |
| 0.6 | 2.88 | 20.03 | 0.88 | 3.91 | 3.48 |
| **0.4 (Ours)** | **2.99** | 19.15 | **0.91** | 4.16 | **3.51** |
| 0.2 | 2.78 | 19.98 | 0.88 | 4.18 | 3.50 |
| 0.0(Diff. Only) | 2.71 | 18.77 | 0.89 | **4.23** | 3.48 |

### E.2. Impact of Sampling Steps ($N$)

We first evaluate the impact of the number of reverse steps $N$ by fixing $T_{rs} = T$ and $\alpha = 0.4$. As shown in Table 9, both perceptual metrics (PESQ, DNS-MOS) and intelligibility (ESTOI) improve significantly as $N$ increases from 10 to 25. However, beyond $N = 25$, the performance gains plateau and even exhibit a slight regression in WV-MOS. This suggests that 25 steps are sufficient to capture the generative distribution of clean speech, and further iterations only increase computational MACs without providing meaningful acoustic refinement.

*Table 9.* Ablation of sampling steps $N$ (fixing $\alpha = 0.4, T_{rs} = T$).

| $N$ | PESQ ↑ | SI-SDR ↑ | ESTOI ↑ | WV-MOS ↑ | DNS-MOS ↑ |
|---|---|---|---|---|---|
| 10 | 2.88 | **20.45** | 0.89 | 3.73 | 3.42 |
| 15 | 2.93 | 21.20 | 0.90 | 3.98 | 3.51 |
| 20 | 2.98 | 21.12 | 0.90 | 4.19 | 3.55 |
| **25** | **3.08** | 20.11 | **0.92** | 4.33 | 3.58 |
| 30 | 3.11 | 19.01 | **0.92** | 4.34 | **3.60** |
| 35 | 3.09 | 19.28 | **0.92** | 4.32 | **3.60** |
| 40 | 3.11 | 18.99 | **0.92** | **4.37** | 3.58 |

### E.3. Impact of Trajectory Starting Point ($T_{rs}$)

We further investigate the effect of the starting time $T_{rs}$ with a fixed discretization step $\Delta t = 0.04$ and $\alpha = 0.4$. As illustrated in Table 10, PESQ continues to show marginal improvements as $T_{rs}$ increases up to 0.24, indicating that a longer diffusion path helps in reconstructing fine-grained spectral details. However, WV-MOS and DNS-MOS reach their peak at $T_{rs} = 0.12$. When $T_{rs} > 0.12$, we observe a gradual decline in these metrics, as the stochastic nature of the diffusion process begins to deviate from the ground-truth waveform. Thus, $T_{rs} = 0.12$ is selected as the optimal starting point to balance generative realism and signal fidelity.

| $T_{rs}$ | PESQ ↑ | SI-SDR ↑ | ESTOI ↑ | WV-MOS ↑ | DNS-MOS ↑ |
|---|---|---|---|---|---|
| 0.04 | 2.61 | **19.45** | 0.87 | 3.76 | 3.36 |
| 0.08 | 2.78 | 19.40 | 0.89 | 3.98 | 3.46 |
| **0.12** | **2.99** | 19.15 | 0.91 | **4.16** | **3.51** |
| 0.16 | 3.06 | 18.89 | 0.91 | 4.11 | 3.48 |
| 0.20 | 3.10 | 18.71 | **0.92** | 4.07 | 3.48 |
| 0.24 | **3.13** | 18.62 | **0.92** | 3.99 | 3.44 |

*Table 10.* Ablation of starting time $T_{rs}$ (fixing $\alpha = 0.4, \Delta t = 0.04$).

## F. Detailed Configuration of Ablation Variants

To further clarify the contribution of each module, we provide the implementation details for the ablation variants discussed in Section 4.7. The architectural modifications are illustrated in Fig. 7.

- **w/o FANC Encoder:** In this variant, the non-uniform spectral partitioning strategy is disabled. Instead of processing the three frequency bands independently with heterogeneous dilated kernels, we employ a standard 2D convolution with a uniform stride of 3 across the entire frequency axis, as shown in Fig. 7(a).

- **w/o FI Module:** The Frequency-Aware Interaction (FI) module is not entirely removed; rather, we deactivate the frequency-selective subpath. In this configuration, the predictive branch provides guidance to the diffusion branch without adaptive band-wise weighting, as depicted in Fig. 7(b).

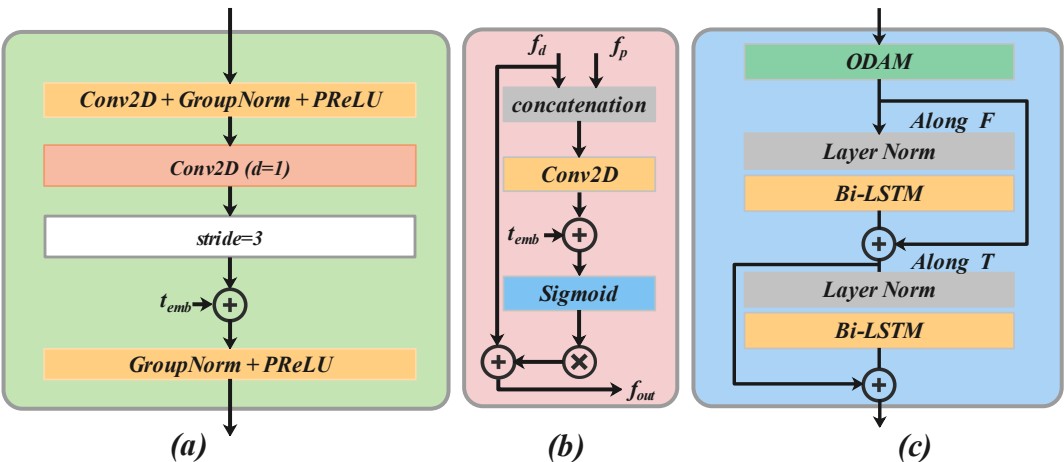

*Figure 7.* Architectural modifications for the ablation study. (a) The simplified encoder replacing FANC, using a uniform stride without band splitting. (b) The FI variant with the frequency-selective branch disabled. (c) The baseline module replacing LISA, featuring a standard dual-path processing structure.

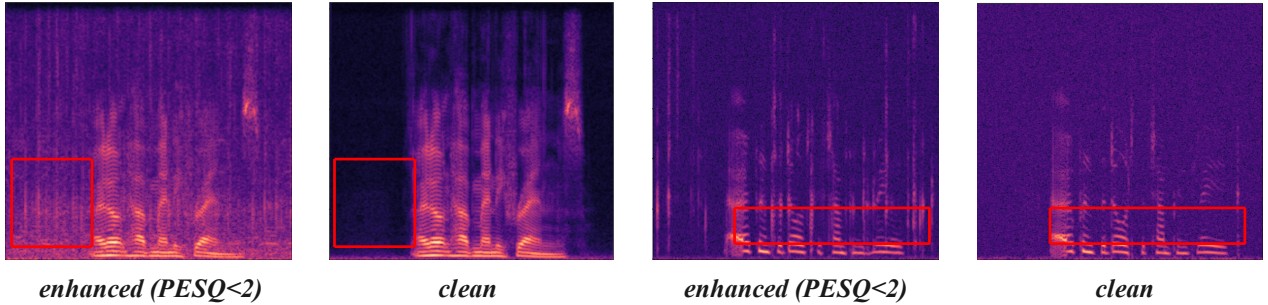

| enhanced (PESQ<2) | clean | enhanced (PESQ<2) | clean |

*Figure 8.* Visualization of two typical artifacts in enhanced speech with PESQ $< 2$. **Left**: High-intensity residual noise localized in low-frequency quasi-silent regions. **Right**: Deficiency of core low-frequency bands caused by the over-suppression of speech energy.

- **w/o Phase Loss:** For this experiment, the anti-wrapping phase regularization term $\mathcal{L}_{\text{pha}}$ is excluded from the optimization objective. The resulting total loss is formulated as:

$$\mathcal{L} = \lambda_1 \mathcal{L}_{\text{mag}} + (1 - \lambda_1)\mathcal{L}_{\text{comp}} + \mathcal{L}_{\text{score}}. \tag{17}$$

- **w/o LISA Module:** The multi-range parallel branches and dynamic filtering mechanism are removed. We replace the LISA module with a conventional dual-path architecture (e.g., intra- and inter-block processing) commonly used in standard speech models (Luo et al., 2020; Zhang et al., 2025), as illustrated in Fig. 7(c).

## G. Performance Analysis of TLB across Quality-based Stratifications

In this section, we provide a detailed rationale for stratifying the test samples based on their baseline PESQ scores and present the experimental validation for our parameter selection. In a standard U-Net architecture, the scaling parameters $(s, b)$ are expected to maintain a delicate equilibrium: $s$ is designed to compensate for spectral energy (particularly in high-frequency details), while $b$ aims to suppress residual noise and preserve low-frequency speech integrity.

### G.1. Case I: Severely Degraded Samples (PESQ $< 2$)

For samples in the lowest quality tier (PESQ $< 2$), the primary challenge lies in rescuing speech information from severe corruption. As illustrated in Fig. 8, the restored spectrograms from the baseline model typically exhibit two critical artifacts:

1. **Over-suppression in Low-frequency Bands:** Since low-frequency components contain the majority of speech energy and phonetic information, the model's tendency to over-denoise leads to a significant loss of intelligibility.

2. **Residual Noise in Quasi-silent Regions:** Significant noise often persists in low-frequency intervals that should ideally be silent, which heavily penalizes the PESQ score.

**Parameter Configuration Logic:** To address these issues, our TLB strategy prioritizes the restoration of low-frequency spectral integrity. We observe that:

- **Low-frequency Enhancement ($s_{1,low}, s_{2,low}$ ↑):** We set these scaling factors to relatively large values. This provides the U-Net with additional high-frequency information and spectral energy in the low-frequency regions to counteract over-suppression.

- **Adaptive Noise Suppression ($b_{1,low}, b_{2,low}$ ↑):** While increasing $s$ restores energy, it inevitably introduces extra noise artifacts. Therefore, the corresponding bias parameters must also be increased to suppress both the inherent residual noise and the artifacts introduced by $s$.

- **The $b < s$ Constraint:** Crucially, we maintain the constraint that the bias values ($b_{1,low}, b_{2,low}$) do not exceed the scaling factors ($s_{1,low}, s_{2,low}$). This ensures that the denoising intensity does not outweigh the energy compensation, resulting in a net gain in speech quality.

In this case, since the overall quality is predominantly dictated by low-frequency restoration, the high-frequency parameters ($s_{high}, b_{high}$) can be assigned lower priority or reduced values to avoid introducing unnecessary high-frequency hiss. To validate this strategy, we conducted an extensive ablation study to identify the optimal balance for severely degraded samples. A representative optimal configuration identified through our grid search is:

- **Low-frequency components:** $s_{1,low} = 2.0, s_{2,low} = 4.1$ and $b_{1,low} = 2.5, b_{2,low} = 1.5$.

- **High-frequency components:** $s_{1,high} = 0.8, s_{2,high} = 0.5$ and $b_{1,high} = 1.0, b_{2,high} = 1.0$.

The performance variations under different parameter combinations are illustrated in Table 11. These results confirm that aggressive scaling in the low-frequency bands, coupled with synchronized bias suppression, yields the most significant PESQ improvements for low-quality speech.

*Table 11.* Ablation study of the four critical low-frequency parameters for Case 1 (PESQ $< 2$).

| Configuration | $s_{1,low}$ | $b_{1,low}$ | $s_{2,low}$ | $b_{2,low}$ | **PESQ** | **STOI** |
|---|---|---|---|---|---|---|
| Baseline (Original) | 1.0 | 1.0 | 1.0 | 1.0 | 1.73 | 0.77 |
| Only Level 1 | 1.5 | 2.0 | 1.0 | 1.0 | 1.75 | 0.77 |
| Only Level 1 | 1.8 | 2.3 | 1.0 | 1.0 | 1.75 | 0.77 |
| Only Level 1 | **2.0** | **2.5** | 1.0 | 1.0 | 1.76 | 0.77 |
| Only Level 1 | 2.2 | 2.7 | 1.0 | 1.0 | 1.75 | 0.77 |
| Only Level 2 | 1.0 | 1.0 | 3.5 | 1.3 | 1.78 | 0.77 |
| Only Level 2 | 1.0 | 1.0 | **4.1** | **1.5** | 1.81 | 0.77 |
| Only Level 2 | 1.0 | 1.0 | 4.5 | 1.7 | 1.73 | 0.77 |
| Only ($s$) | 2.0 | 1.0 | 4.1 | 1.0 | 1.61 | 0.76 |
| Only ($b$) | 1.0 | 2.5 | 1.0 | 1.5 | 1.71 | 0.77 |
| **Optimal TLB** | **2.0** | **2.5** | **4.1** | **1.5** | **1.88** | **0.78** |

### G.2. Case II: Moderately Restored Samples ($2 <$ PESQ $\leq 3$)

In the second tier, the restored speech generally exhibits high fidelity in the low-frequency regions with minimal structural degradation, contributing to a relatively high baseline PESQ. However, the primary remaining issue is the presence of subtle, quasi-uniform residual noise distributed across the entire spectro-temporal domain.

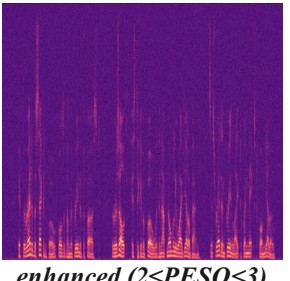 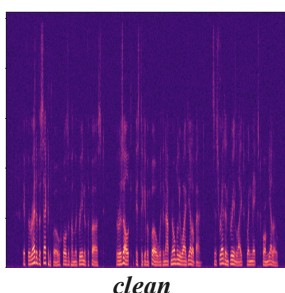 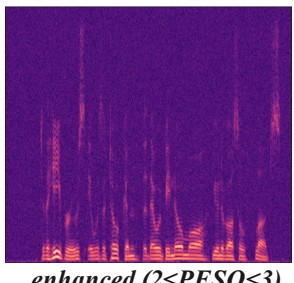 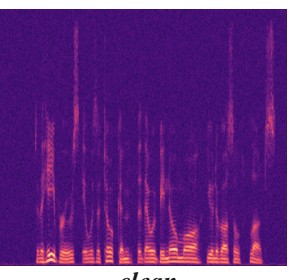

| *enhanced (2<PESQ<3)* | *clean* | *enhanced (2<PESQ<3)* | *clean* |

*Figure 9.* Visualization of two typical artifacts in enhanced speech with $2 < \text{PESQ} \leq 3$.

**Parameter Configuration Logic:** Given that the artifacts are no longer concentrated in specific frequency bands, our strategy shifts toward global consistency. The parameter configuration follows these principles:

- **Uniform Parameter Application:** To address the uniform distribution of noise, we synchronize the parameters across both high and low frequency bands, such that $s_{high} \approx s_{low}$ and $b_{high} \approx b_{low}$. This ensures a balanced restoration without introducing spectral tilts.

- **Aggressive Noise Suppression** ($b \uparrow$)**:** The bias parameters are set to relatively large values to enhance the model's ability to suppress the persistent background hiss and "clear up" the global spectrogram.

- **Energy Compensation** ($s > 1$)**:** Concurrently, the scaling factors ($s$) are also set greater than 1. This acts as a protective mechanism to prevent the aggressive bias term ($b$) from over-purifying the signal, which could otherwise lead to the loss of subtle speech components or "thinning" of the audio.

- **The $s < b$ Balance:** To ensure that the strategy remains effective for denoising, we maintain the constraint $s < b$. In this specific tier, this relationship ensures that the noise reduction intensity slightly outweighs the energy amplification, effectively lowering the noise floor without compromising the already well-restored speech structure.

In light of these principles, we adopt a frequency-agnostic configuration for this tier, applying identical parameters to both low and high-frequency bands to maintain spectral balance. Specifically, the optimized parameters are set as follows:

- **Level 1:** $s_{1,low} = s_{1,high} = 1.5$ and $b_{1,low} = b_{1,high} = 2.5$.

- **Level 2:** $s_{2,low} = s_{2,high} = 2.0$ and $b_{2,low} = b_{2,high} = 1.5$.

This configuration reflects our strategy of aggressive noise suppression in the first stage ($b_1 > s_1$), followed by controlled energy recovery in the second stage ($s_2 > b_2$). To validate the efficacy of this cross-band uniform application and the selection of these specific values, we conducted a targeted ablation study, as summarized in Table 12.

*Table 12.* Ablation study of global uniform parameters for Case 2 ($2 < \text{PESQ} \leq 3$). All parameters are synchronized across low and high frequencies ($s_{low} = s_{high}, b_{low} = b_{high}$).

| Configuration | $s_1$ | $b_1$ | $s_2$ | $b_2$ | PESQ | STOI |
|---|---|---|---|---|---|---|
| Baseline (Original) | 1.0 | 1.0 | 1.0 | 1.0 | 2.59 | 0.82 |
| Level 1 Only | 1.5 | 2.5 | 1.0 | 1.0 | 2.61 | 0.82 |
| Level 2 Only | 1.0 | 1.0 | 2.0 | 1.5 | 2.62 | 0.82 |
| **Uniform TLB (Optimal)** | **1.5\*** | **2.5\*** | **2.0\*** | **1.5\*** | **2.65** | **0.82** |

*The optimal parameters ($s_1, b_1, s_2, b_2$) were identified through an exhaustive grid search and comprehensive empirical validation to ensure robust performance.

## G.3. Case III: High-Fidelity Samples (PESQ $\geq 3$)

For samples with PESQ $\geq 3$, the restored speech already exhibits high fidelity, characterized by nearly intact spectral structures and negligible residual noise. In this high-quality regime, the focus of restoration shifts from fundamental reconstruction to the refinement of fine-grained spectral details.

**Dominance of High-Frequency Components:** We posit that for high-fidelity speech, the high-frequency regions play a decisive role in further improving perceptual quality. This is because:

- **Low-frequency Saturation:** At this quality level, the low-frequency backbone of the speech is typically recovered to a near-optimal state. Continued intervention in the low-frequency bands yields diminishing returns and poses a risk of introducing unnecessary artifacts into an already stable signal.

- **Perceptual Brilliance:** The subjective clarity and naturalness of speech are highly sensitive to high-frequency harmonics and overtones. Minor spectral dampening or fine-grained noise in the high-frequency range becomes the primary bottleneck preventing the score from reaching a near-perfect level (e.g., PESQ ¿ 4).

**Parameter Configuration Logic:** Based on the above rationale, we implement a selective optimization strategy:

- **High-frequency Specific Tuning:** We exclusively apply the TLB strategy to the high-frequency scaling and bias parameters ($s_{high}, b_{high}$). By fine-tuning these values, we can subtly enhance the high-frequency energy and sharpen the spectral peaks, thereby improving the overall "crispness" of the audio.

- **Low-frequency Preservation:** The low-frequency parameters ($s_{low}, b_{low}$) are maintained at their identity values (i.e., $s = 1, b = 0$). This "hands-off" approach ensures that the high-quality low-frequency components, which are already well-restored by the U-Net, remain untouched, preserving the fundamental integrity of the speech.

Given that the baseline quality in this tier is already near-optimal, the numerical improvement afforded by the TLB strategy is relatively marginal (approximately 0.02 PESQ). Consequently, visual inspection of the spectrograms yields no discernible differences; thus, we omit the visual comparison for the sake of brevity. However, to achieve this subtle refinement in perceptual clarity, we identified the following specific parameter configuration:

- **Low-frequency components (Preservation):** $s_{1,low} = 1.0, s_{2,low} = 1.0$ and $b_{1,low} = 1.0, b_{2,low} = 1.0$.

- **High-frequency components (Refinement):** $s_{1,high} = 1.1, s_{2,high} = 1.2$ and $b_{1,high} = 1.2, b_{2,high} = 1.1$.

In this configuration, the low-frequency parameters are kept at their identity mapping to safeguard the fundamental speech structure, while the high-frequency parameters are slightly elevated to enhance spectral brilliance. This selective boosting confirms that for high-fidelity audio, "less is more," and targeted high-frequency adjustment is sufficient to capture the remaining perceptual gains.

# H. Visual results

To provide a more intuitive demonstration of the effectiveness of the TLB strategy, we provide a comparative visualization of the spectrograms in Fig. 10. The red-boxed regions highlight the distinct differences between the clean reference, the baseline enhanced output, and the TLB result.

It is observable that under conditions of severe degradation, the baseline enhancement model often fails to completely eliminate residual noise, which manifests as prominent *vertical spectral streaks* (as shown in the red boxes). These artifacts represent structured noise that heavily degrades the perceptual quality and intelligibility. Upon applying the TLB strategy, these vertical structures are significantly attenuated or, in some cases, entirely removed. This qualitative refinement confirms that TLB not only improves numerical metrics but also effectively "cleans" the spectro-temporal regions that are most challenging for standard U-Net architectures, leading to a substantial gain in overall speech quality.

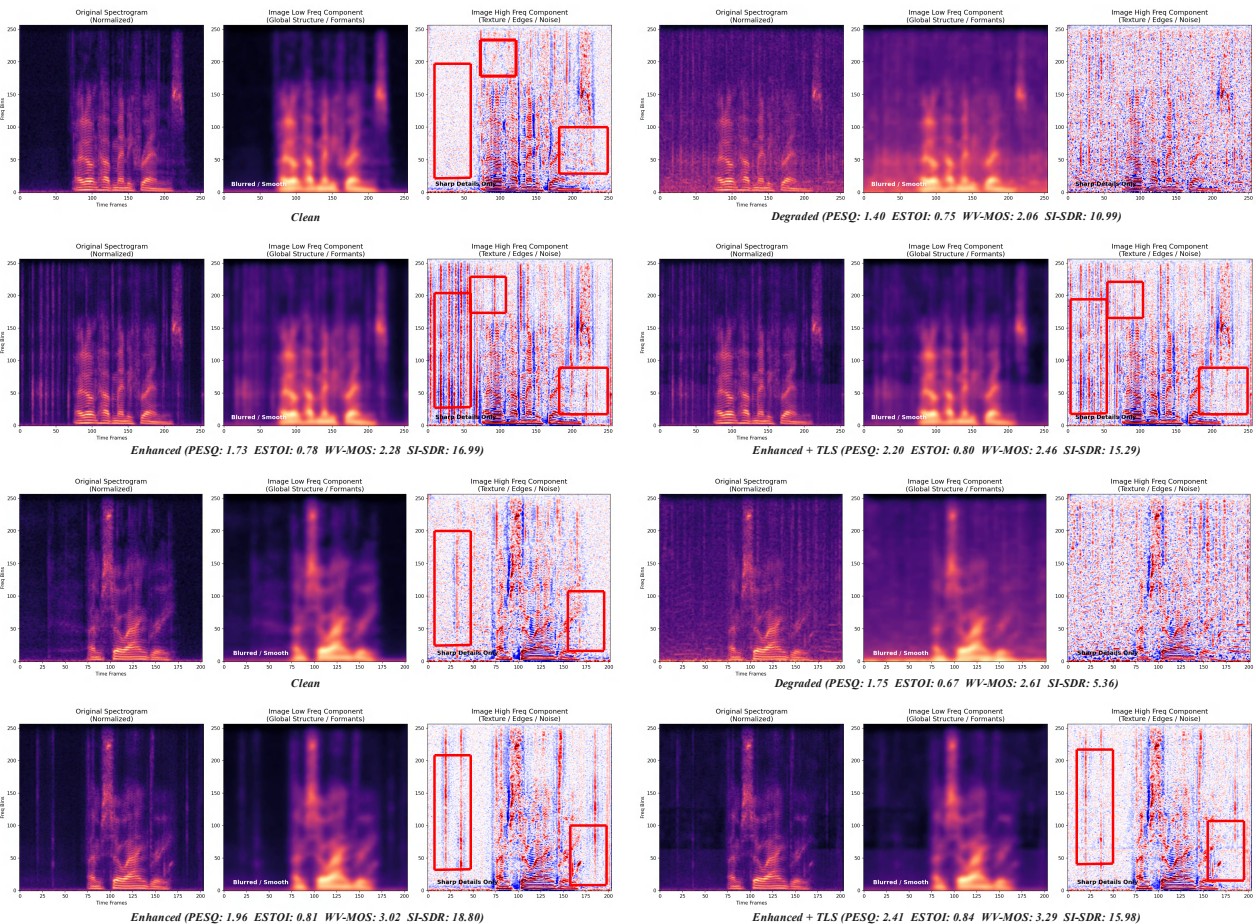

*Figure 10.* Visual comparison of spectrograms for clean speech, baseline enhanced speech, and speech refined by the TLB strategy. The red boxes highlight regions where the baseline model fails to suppress high-intensity noise streaks, whereas our TLB strategy effectively attenuates or eliminates these artifacts.

