# OpenReview forum: "Dual-View Predictive Diffusion: Lightweight Speech Enhancement via Spectrogram-Image Synergy"
_ICML.cc/2026/Conference — ICML 2026 regular_

### Official Review · Reviewer_YZH6 · 2026-02-24

**Soundness:** 3
**Presentation:** 3
**Significance:** 2
**Originality:** 3
**Overall Recommendation:** 4
**Confidence:** 4

**Summary:**

This paper proposes a dual-branch framework that combines a predictive branch for coarse magnitude and phase with a diffusion branch for detail refinement. To improve efficiency, it introduces several lightweight modules (FANC, FI, LISA) and an inference-time strategy (TLB). Experiments on WSJ0-UNI and VoiceBank+DEMAND show quality and efficiency results.

**Compliance With Llm Reviewing Policy:**

Affirmed.

**Final Justification:**

Thank you for your rebuttal. The rebuttal answers my major concerns on formulation and robustness. I will raise my score to 4.

**Key Questions For Authors:**

1. How is TLB tier selection performed at inference time without access to reference-based PESQ?
2. What is the exact hyperparameter protocol relating Trs, N, and Δt across all inference-time studies?
3. Can you report real-time performance (RTF/latency) and provide module-level cost breakdowns?
4. Is Spectrogram-Image Synergy the main contribution, and if so, can you re-position Related Work and the main text accordingly?

**Limitations:**

No.
See Weaknesses.

**Strengths And Weaknesses:**

Strengths:
1. The module designs actually make sense.
2. The evaluation is fairly thorough. They test on multiple benchmarks with different metrics, not just one.
3. The paper organization is clear.

Weaknesses:

1. The motivation is just not clear. The paper points out that predictive models have problems and diffusion models might help, but it never really explains why the proposed architecture has to be diffusion-based. Why couldn't these ideas work with other types of models? The only thing diffusion seems to add is "better detail recovery," which is too vague. So at the end, I am still not sure what is actually new here and why it needs to be done this way.

2. The paper spends way too much time explaining diffusion models. There is a whole section in related work, more background before the method, and even an appendix on diffusion theory. It reads like a diffusion tutorial. But the actual novel idea is about looking at spectrograms from two angles, acoustic structure and image texture. That part is mentioned but never really discussed or compared to what others have done. The main contribution just gets buried under all the diffusion stuff.

3. The whole tiering thing based on PESQ does not work in the real world. First, the thresholds 2 and 3 look arbitrary, no data to back them up. Second, PESQ needs a clean reference to compute, but in practice you never have that. So how would anyone actually assign a sample to a tier during inference? The paper never explains that. And on top of that, the optimal parameters for each tier come from exhaustive search, which means the method is really an offline analysis tool, not something you can actually deploy.

4. The inference settings are not consistent across experiments. The paper first gives a default setting (Trs = 0.12, N = 3). But later, when testing the effect of N, it changes Trs to T instead of keeping it at 0.12. When testing Trs, it fixes Δt = 0.04. The issue is that Δt and N are mathematically linked (Δt = T/N), so changing these settings without a clear rule makes it unclear what is actually causing the performance difference. I cannot understand whether improvements come from fewer steps, a shorter diffusion range, or different step sizes.

5. The paper keeps saying the method is lightweight, but never actually proves it. We get MACs and parameter counts, which is fine, but no real-time factor on any actual hardware. So how do we know if this can run in practice? The authors also never break down how much each module (FANC, LISA) costs, so we have no idea what is actually expensive. They claim TLB adds no overhead during inference, but again, no timing numbers to back that up. And while they experiment with different N and Trs values, they never show how real-time factor changes when you adjust these.

---

> ### Author Rebuttal · Authors · 2026-03-29
>
> We sincerely thank the reviewer for these important concerns and suggestions.
>
> **Regarding weakness 1**
>
> Predictive and diffusion models represent two fundamental paradigms in speech enhancement. As discussed in Introduction, diffusion models are particularly effective at recovering fine spectral structures under severe degradation or distribution mismatch, but they suffer from much higher inference cost than predictive models.
>
> To leverage their complementary strengths, we adopt a **hybrid** design in the proposed framework: the predictive branch first produces a coarse estimate, while the diffusion branch refines ambiguous details. In this way, the proposed method combines the efficiency of predictive models with the refinement capability of diffusion models, enabling a lightweight yet robust enhancement framework. We will revise the manuscript to clarify this motivation and design rationale.
>
> **Regarding weakness 2**
>
> We agree that the diffusion part should not occupy too much space or include excessive theoretical details. In the revision, we will substantially shorten this diffusion part in both the main text and appendix.
>
> The actual novelty of our work lies in a **lightweight and efficient framework built on Spectrogram-Image Synergy**, i.e., modeling spectrograms jointly as acoustic structure and image texture. This principle directly motivates **FANC** and **LISA**, which form the core architectural contributions of **DVPD**. The diffusion-based methods discussed in current **Related Work** are generally designed from only a single perspective, whereas our method explicitly combines both views. Table 1 further shows that **DVPD** consistently outperforms these prior methods. We will re-organize the Related Work accordingly, highlighting these prior methods from their respective single-view perspectives and making our dual-view design much more explicit.
>
> **Regarding weakness 3**
>
> We clarify that thresholds **2** and **3** were not arbitrary. In speech enhancement field, **PESQ < 2** is commonly associated with **severely degraded** speech, while **PESQ > 3** usually corresponds to **high-quality** speech. For instance, the former often contains distorted low-frequency structure, whereas the latter is typically already perceptually clean.
>
> In practice, the PESQ tier assignment can be replaced by a reference-free estimator such as **DNSMOS**.
>
> |Selector|Low|Mid|High|PESQ|DNSMOS|
> |-|-|-|-|-|-|
> |PESQ|<2|2–3|>3|3.15|3.67|
> |DNSMOS|<2.5|2.5–3.5|>3.5|3.13|3.63|
>
> These results show that the PESQ-tiered can be translated into a practical deployment.
>
> TLB is **online** deployable due to the following reason: once the optimal parameters are determined offline for each tier, no further optimization is required online. The online stage only needs to select the appropriate tier, which can be done using DNSMOS to estimate input quality and choose the corresponding TLB branch. We will clarify this **offline–online** distinction in the revision.
>
> **Regarding weakness 4**
>
> Our studies use a **two-stage protocol**. First, we fix $T_{rs}=T$ and vary $N$ (so $\Delta t=T/N$) to choose a reasonable step size, table 8 shows that gains become marginal once $N>25$, so we use $\Delta t\approx0.04$. Second, after fixing $\Delta t=0.04$, we vary $T_{rs}$ (equivalently, $N=T_{rs}/\Delta t$) to determine the minimal effective reverse range (Table 9). Based on above, we set the default settings **$N = 3$** and **$T_{rs} = 0.12$** in the paper.
>
> **Regarding weakness 5**
>
> We conducted runtime experiments on hardware. Default setting: **$N = 3$**, **$T_{rs} = 0.12$**. **RTF** denotes the inference time (s) required to process 1s audio.
>
> |GPU|Model|Params(M)|MACs(G)|RTF|
> |-|-|-|-|-|
> |A6000|DVPD-P|0.61|2.41|0.0097|
> |A6000|PGUSE|5.1|26.3|0.1012|
> |A6000|DVPD|1.9|10.2G|0.0475|
>
> These results show that **DVPD** is faster than **PGUSE**, while **DVPD-P** is faster still.
>
> We also provide a module-level cost breakdown:
>
> |Module|Params(M)|MACs(G)|
> |-|-|-|
> |FANC encoder|0.06|0.32|
> |FANC decoder|0.07|0.45|
> |LISA(C=24)|0.03|0.27|
> |LISA(C=48)|0.09|0.24|
> |LISA(C=96)|0.28|0.23|
>
> For **TLB**, it is a **training-free** strategy.
>
> |TLB|RT|
> |-|-|
> |without|0.0475|
> |offline|0.0475|
> |online|0.10~0.12|
>
> We fix $\Delta t = 0.04$, measure the RTF under different steps:
>
> |$N$($T_{rs}$)|RTF|
> |-|-|
> |2(0.08)|0.0342|
> |3(0.12)|0.0475|
> |4(0.16)|0.0581|
> |6(0.24)|0.0994|
>
> These results make the runtime-quality tradeoff more explicit.
>
> **Regarding question 1**
>
> In practical deployment, PESQ-based tier can be replaced by **DNSMOS**. More details are in **weakness 3**.
>
> **Regarding question 2**
>
> The default setting is **$N = 3$**, **$\Delta t = 0.04$**, and **$T_{rs} = 0.12$**. More details are in **weakness 4**.
>
> **Regarding question 3**
>
> Yes. We additionally provide runtime results. More details are in **weakness 5**.
>
> **Regarding question 4**
>
> Yes. We will re-position related work and the main text. More details are in **weakness 2**.

---

> > ### Author Rebuttal · Reviewer_YZH6 · 2026-04-02
> >
> > Thank you for the rebuttal. The additional explanations are helpful, and I appreciate that the authors responded directly to several of my practical concerns, especially regarding the runtime measurements, the default settings for \(N\), \(\Delta t\), and \(T_{rs}\), and the intended role of TLB. That said, I still feel that some of the central issues are only partially resolved. My main remaining concern is about the overall positioning of the contribution. The rebuttal makes it clearer that the intended novelty is the lightweight dual-view design built around spectrogram-image synergy, with FANC and LISA as the key architectural components, while the diffusion part serves more as a refinement mechanism. However, this also reinforces my earlier impression that the paper itself does not yet present this contribution with sufficient clarity. In its current form, the manuscript still spends substantial space motivating diffusion and hybrid predictive-diffusion design in a fairly general way, whereas the actual distinctive idea seems to be the dual-view treatment of the spectrogram and the corresponding architectural choices. I therefore still think the authors need to sharpen the narrative much more clearly, so that the reader can understand what the principal contribution really is and why the diffusion-based formulation is specifically needed for this contribution, rather than merely being one possible implementation choice.
> >
> > I also remain somewhat unconvinced about the practical story around TLB. The rebuttal is useful in clarifying that the tier parameters are determined offline and that DNSMOS can be used as a reference-free proxy at deployment time, but this still leaves an important gap for me. In particular, the practicality of the online version depends not only on whether a proxy exists, but also on whether the full deployment procedure with that proxy has been validated end-to-end. At the moment, it still feels as though the paper’s main evidence for TLB is tied to a quality-tiering logic originally defined using PESQ, while the rebuttal proposes DNSMOS as a plausible substitute rather than demonstrating that the same conclusions remain valid under the actual reference-free setting. Relatedly, while I appreciate the newly provided runtime numbers, they also make me want a more careful explanation of the claim that TLB adds essentially no practical overhead, since the reported online runtime appears meaningfully higher than the base model. This does not necessarily invalidate the method, but it does suggest that the paper should be much more precise about what kind of overhead is negligible, under what implementation setting, and whether the trade-off is between quality and latency rather than “free” improvement. More broadly, although the response on \(N\), \(\Delta t\), and \(T_{rs}\) is much better than before, I still think the manuscript would benefit from presenting this protocol more transparently, because these parameters are mathematically coupled and the current discussion still feels somewhat procedural rather than principled. Finally, the added module-level cost breakdown is helpful, but it is still closer to a static complexity summary than a full explanation of what actually dominates runtime in practice. Overall, I do think the rebuttal improves my understanding of the work, but I still believe the paper would need clearer contribution positioning, a more convincing deployment-oriented validation of TLB, and a more precise treatment of the quality-latency trade-off before the practical claims are fully supported.

---

> > > ### Author Response · Authors · 2026-04-03
> > >
> > > **Regarding the overall positioning of the contribution**
> > >
> > > Since the rebuttal stage does not allow manuscript revision, we briefly clarify the motivation and contribution.
> > > Our goal is to design a lightweight, efficient speech enhancement framework. The core idea to realize our goal is the dual-view spectrogram modeling, which jointly models spectrograms from acoustic structure and image texture perspectives. Image-based methods are effective for texture modeling but often ignore acoustic structure, while acoustic-based methods capture frequency-dependent structure but underutilize image-style representations. Our dual-view design aims to combine the strengths of both.
> > >
> > > Diffusion models are well suited for modeling 2D texture distributions through iterative refinement, which aligns naturally with our image-view modeling. However, diffusion alone is computationally expensive. Therefore, we adopt a hybrid design with a clear division of roles: the predictive branch produces a coarse estimate to capture global structure, while the diffusion branch refines residual texture details. This hybrid design allows the model to better exploit the image-view representation while enabling a better quality–efficiency trade-off. In the revision, we will reorganize the paper to clarify the dual-view architecture is the **core contribution**, while FANC, LISA, and the predictive–diffusion hybrid design are components that realize this idea.
> > >
> > > **Regarding end-to-end deployment with DNSMOS-based tiering**
> > >
> > > In previous rebuttal, we already replaced PESQ-based tiering with DNSMOS-based tiering and conducted an end-to-end evaluation. In the table(Weakness 3), the “PESQ” column (3.15 for PESQ-tiering and 3.13 for DNSMOS-tiering) reports the final PESQ scores after end-to-end inference. This shows that **DNSMOS-based** tiering still provides a substantial improvement compared to 2.99 without any tiering and achieves a very similar gain to PESQ-tiering. This result is consolidated when similar results are observed using DNSMOS score (3.63 vs 3.67).
> > >
> > > In this rebuttal, we keep the PESQ-based offline parameters and only replace the tier with DNSMOS, which already yields comparable results. But we agree a fully **DNSMOS-native tiering system** with reoptimized tier boundaries and parameters would be more complete.  We will include a fully DNSMOS-based re-calibration in the revision.
> > >
> > > **Regarding the practical overhead and latency trade-off of online TLB**
> > >
> > > Based on runtime results, the online version of TLB does introduce additional latency: the runtime increases from 0.0475 to 0.10–0.12, mainly because DNSMOS must be executed for tier assignment. In the online setting, TLB should be understood as a quality–latency trade-off, rather than a “free” improvement.
> > >
> > > What we intended to emphasize is : TLB is **training-free** and does not introduce any additional training cost or retraining procedure, since TLB only adjusts the parameters of **“low-frequency/high-frequency components”** and uses a pretrained DNSMOS model for tier selection. In this sense, the “no overhead” claim applies to the training stage, not the online inference stage.We will clarify this in the revision.
> > >
> > > **Regarding the principled interpretation of N , ∆t, and Trs**
> > >
> > > We determine N and Trs motivated by the diffusion reverse process.**Stage 1:** The reverse process can be viewed as a discretized trajectory from T to 0, where ∆t controls the discretization granularity and N is the number of reverse steps. We vary N over the full reverse trajectory (Table 8) to identify a proper discretization level. The results show that performance saturates when N>25,indicating that ∆t=0.04 is sufficiently fine.**Stage 2:** We then adopt truncated diffusion strategy (such as PGUSE), where the reverse process starts from a pre-generated estimate (predictive output here) instead of pure noise. The key question becomes which diffusion stage this estimate corresponds to, we denote this truncation starting point as Trs. By fixing ∆t=0.04 and varying Trs (Table 9), the best performance is achieved at Trs=0.12, corresponding to N=Trs/∆t=3. This suggests that the predictive branch output approximately corresponds to the diffusion state at **T=0.12**, so earlier reverse steps can be skipped to reduce complexity. We use N=3 and Trs=0.12 based on this.
> > >
> > > **Regarding what dominates runtime in practice**
> > >
> > > We further profiled the runtime to identify the practical bottleneck. Out of 47.5ms total, the predictive branch takes 11.0ms, while the diffusion branch (N=3) takes 36.5ms (≈12.1ms per step),indicating that the diffusion branch dominates runtime.
> > > Per diffusion step, the breakdown is: FANC encoder (0.74ms), FANC decoder (0.26ms), two LISA (C=24) (2.9ms each), two LISA (C=48) (1.1ms each), and a LISA (C=96) (0.7ms), totaling 9.7ms out of 12.1ms, showing that most runtime comes from LISA.This matches our design, where LISA is the main performance-contributing component and also the main runtime cost.

---

### Official Review · Reviewer_wXgQ · 2026-03-11

**Soundness:** 3
**Presentation:** 3
**Significance:** 3
**Originality:** 3
**Overall Recommendation:** 4
**Confidence:** 4

**Summary:**

Overall, this study's principal objective is to build an extremely lightweight speech enhancement model by exploiting the dual nature of spectrograms—simultaneously treating them as physical frequency-domain representations and as 2D visual textures—rather than applying spatially uniform operations that ignore frequency-domain structure.

The paper presents DVPD (Dual-View Predictive Diffusion), a parallel predictive-diffusion hybrid for universal speech enhancement (USE). The paper's key result is that DVPD outperforms significantly larger models across multiple benchmarks while achieving a new efficiency frontier for generative speech enhancement.

**Compliance With Llm Reviewing Policy:**

Affirmed.

**Key Questions For Authors:**

**ODAM isolation**: Could you add an ablation replacing ODAM with a standard attention module (e.g., multi-head self-attention) while keeping LISA? This would clarify how much performance is attributable to LISA specifically vs. the unpublished ODAM component.

**Limitations:**

Yes

**Strengths And Weaknesses:**

### Strengths

**Efficiency gains are real and substantial.** The reported numbers speak clearly: 1.9M vs 5.1M params (PGUSE), 10.2G vs 26.3G MACs, with better PESQ and WV-MOS across the board. DVPD-P alone (0.61M, 2.41G MACs) matches predictive-only SOTA MP-SENet (2.26M, 34.58G MACs)—a 14× MACs reduction with comparable quality.

**LISA is the standout contribution.** The ablation in Table 4 is clear: removing LISA drops PESQ from 2.99 to 2.71 and WV-MOS from 4.16 to 3.85—by far the largest single-component degradation. Stripe dynamic convolutions with anisotropic dilations for separately modeling horizontal harmonics and vertical transients are well-motivated by the acoustic physics of spectrograms.

**Thorough evaluation.** Six benchmarks covering universal SE, single-distortion denoising, speech super-resolution, and zero-shot OOD generalization. Fig. 5 is particularly convincing—DVPD consistently dominates across unseen datasets despite training only on WSJ0-UNI.

### Weaknesses

**TLB generalizability is not tested.** TLB modulates U-Net skip connections and backbone features during the reverse diffusion process—a technique that is, in principle, applicable to any U-Net-based diffusion model. The paper compares "DVPD+TLB" against baselines without TLB (PGUSE, StoRM, SGMSE+, etc.). Since TLB is training-free and not architecturally specific to DVPD, applying an equivalent strategy to baseline models would yield a fairer comparison. Could PGUSE or StoRM also benefit from a similar inference-time scaling, potentially closing the gap? If TLB is truly universal, the gain it shows here partly reflects an unfair experimental setup rather than architectural superiority.

**TLB practicality is questionable.** The 8 scaling parameters are tuned separately for three PESQ-based quality tiers. In deployment, PESQ is not available without a clean reference signal. Without a reference-free quality estimator for tier selection, TLB's benefit is hard to realize in practice.

---

> ### Author Rebuttal · Authors · 2026-03-28
>
> We sincerely thank the reviewer for these important concerns and suggestions.
>
> **Regarding weakness 1**
>
> This is an important point that was not sufficiently evaluated in the original submission. As discussed in the paper, the motivation of **TLB** is not specific to DVPD itself, but to a broader property of **U-Net-based diffusion backbones**: during reverse diffusion, they tend to restore low-frequency structures relatively well while being more conservative in high-frequency regions. From this perspective, TLB should, in principle, be applicable to other U-Net-based diffusion models as well.
>
> To directly address this concern, we applied **TLB** to two representative baselines on **WSJ0-UNI**. To also address the practicality issue in Weakness-2, we use **DNSMOS-based tier selection** instead of PESQ. For **StoRM**, which contains only a diffusion branch, we apply TLB only to the first two U-Net stages, following the same setting as DVPD. For **PGUSE**, since its original implementation is not strictly U-Net-shaped, we construct a **U-Net variant of PGUSE** while keeping the remaining modules unchanged, and again apply TLB only to the first two stages of the diffusion branch.
>
> | Model | Tier selector | PESQ | DNSMOS |
> |---|---|---:|---:|
> | StoRM | None | 2.75 | 3.11 |
> | StoRM + TLB | DNSMOS | 2.81 | 3.13 |
> | PGUSE (U-Net) | None | 2.82 | 3.22 |
> | PGUSE (U-Net) + TLB | DNSMOS | 3.02 | 3.55 |
> | DVPD | None | 2.99 | 3.47 |
> | DVPD + TLB | DNSMOS | 3.13 | 3.63 |
>
> These results support three observations.
> 1. **TLB is a transferable plug-in strategy for U-Net-based diffusion models**, even when PESQ-based tiering is replaced by practical **DNSMOS-based** tiering.
>
> 2. **PGUSE (U-Net)** shows the largest gain after applying TLB. A plausible reason is that its weaker baseline causes more samples to fall into the low-quality tier, where TLB provides the strongest improvement. Even so, its final performance is still below **DVPD + TLB**.
>
> 3. **StoRM** shows the smallest gain. One possible explanation is that its deeper U-Net backbone weakens the relative effect of modulation applied only at the first two stages. We present this as a tentative interpretation.
>
> Overall, these experiments support that **(i)** TLB is a general training-free inference booster for U-Net-based diffusion models, and **(ii)** the advantage of DVPD is not solely due to TLB, since DVPD still achieves the best final performance under this fairer comparison.
>
> **Regarding weakness 2**
>
> In practice, the PESQ can be replaced by a **reference-free speech quality estimator**, such as **DNSMOS** or **WV-MOS**.
>
> Using **DNSMOS** as an example, we analyzed its correspondence with PESQ on both **WSJ0-UNI** and **VBDMD**. We found that when **DNSMOS < 2.75**, the great majority of samples fall into the **PESQ < 2** tier, while when **2.75 < DNSMOS < 3.5**, most samples fall into the **2 < PESQ < 3** tier. Since the lowest-quality samples are also the regime where TLB tends to provide the largest gain, we adopt a slightly more conservative deployment rule:
> - **DNSMOS < 2.5** $\rightarrow$ **PESQ < 2**
> - **2.5 < DNSMOS < 3.5** $\rightarrow$ **2 < PESQ < 3**
> - **DNSMOS > 3.5** $\rightarrow$ **PESQ > 3**
>
> These results suggest that the current PESQ-tiered TLB can be translated into a practical deployment strategy using a no-reference quality predictor, without requiring access to clean speech. We will clarify in the revision that the current PESQ-based setting should be viewed as an oracle analysis, while DNSMOS-based tier selection provides a practical reference-free implementation path.
>
> **Regarding question 1**
>
> Following your suggestion, we replaced **ODAM** with a standard **multi-head self-attention (MHA)** module while keeping **LISA** and all other components unchanged.
>
> We observe that using **ODAM** instead of **MHA** improves performance and computational complexity, especially as the inference length becomes longer. This verifies the advantage of **ODAM** over **MHA**.
>
> At the same time, regardless of whether MHA or ODAM is used, the performance of LISA remains largely unchanged. In contrast, as shown in Table 4, removing LISA leads to a substantial performance drop. This observation suggests that the performance improvement of DVPD is mainly attributed to the LISA module rather than ODAM.
>
> | Model variant | PESQ | MACs (1s) | MACs (2s) |
> |---|---:|---:|---:|
> | DVPD (with LISA + ODAM) | 2.9913 | 10.2G | 20.3G |
> | DVPD (with LISA + MHA) | 2.9841 | 10.6G | 23.7G |

---

> > ### Author Rebuttal · Reviewer_wXgQ · 2026-04-04
> >
> > Thanks for the response and I will keep my score.

---

### Official Review · Reviewer_QAde · 2026-03-13

**Soundness:** 3
**Presentation:** 3
**Significance:** 3
**Originality:** 3
**Overall Recommendation:** 4
**Confidence:** 4

**Summary:**

This paper proposes a parallel predictive-and-diffusion‑based speech enhancement network by explicitly exploiting the dual nature of spectrograms as both visual images and acoustic frequency‑domain representations. Specifically, it integrates three main modules as contributions: FANC (Frequency‑Adaptive Non‑uniform Compression), LISA (Lightweight Image‑based Spectro‑Awareness) and TLB (Training‑free Lossless Boost). The model is evaluated extensively on WSJ0‑UNI, VoiceBank+DEMAND, reverberant, bandwidth expansion, and multiple out‑of‑distribution scenarios, achieving consistent gains across several quality metrics.

**Compliance With Llm Reviewing Policy:**

Affirmed.

**Final Justification:**

The concerns have been addressed. Considering the contribution of this paper, I will keep my original positive rating 4 - weak accept.

**Key Questions For Authors:**

1) For VoiceBank+DEMAND, the predictive variant DVPD-P is far from state-of-the-art. This raises the concern that the FANC encoder is not efficient.
2) It is helpful to include WER numbers for speech intelligibility measure as well.
3) For FANC, the non‑uniform frequency compression is one of the paper’s strongest ideas, but the conception is not new. It is better to compare with other non-uniform frequency compression techniques to demonstrate the effectiveness of current design.
4) How sensitive is DVPD to the frequency partition thresholds in FANC and TLB (e.g., 2 kHz)?

**Limitations:**

Some failure case study would be helpful.

**Strengths And Weaknesses:**

Strengths:
1) Strong and well‑motivated core insight. Treating spectrograms as both images and physical frequency representations is not new conceptually, but DVPD clearly and systematically instantiates this idea. In addition, the design of training-free TLB block looks interesting.
2) Good efficiency-quality tradeoff. DVPD reduces MACs by an order of magnitude relative to classic score‑based models with better quality.
3) Thorough experiments across various tasks show good performance.

Weaknesses:
1) For VoiceBank+DEMAND, the proposed is far from state-of-the-art, where the predictive variant DVPD-P achieves a PESQ of 3.14 versus 3.50 by MP-SENet.
2) Some parameters are highly heuristic, e.g. TLB depends on manually tuned scaling parameters and frequency thresholds (e.g., 2 kHz split). This raises questions about the robustness across sampling rates or languages.

---

> ### Author Rebuttal · Authors · 2026-03-28
>
> We sincerely thank the reviewer for these important concerns and suggestions.
>
> **Regarding weakness 1**
>
> **MP-SENet** is stronger on VoiceBank+DEMAND, but **DVPD-P** is not intended to surpass such predictive SOTA models on this dataset. Instead, **DVPD-P** is designed as an **extremely lightweight predictive branch** for resource-constrained settings. Although MP-SENet achieves higher PESQ (3.50 vs. 3.14), it uses **2.26M parameters** and **34.58G MACs**, compared with only **0.61M parameters** and **2.41G MACs** for DVPD-P. DVPD-P is only about 7% of that of MP-SENet. Moreover, Fig. 5 also shows that DVPD-P exhibits better generalization than MP-SENet on some datasets.
>
> **Regarding weakness 2**
>
> The **2 kHz** is not chosen arbitrarily, but is motivated by the non-uniform acoustic structure of speech: the **0–2 kHz** region contains substantial pitch/formant and intelligibility-related cues, while higher-frequency regions are sparser and mainly contribute finer details. The **TLB scaling parameters** do involve empirical tuning, but their ranges were first determined by the dual-view design principle and then refined through validation (Appendix G).
>
> Regarding **sampling rates**, while 0–2 kHz remains physically meaningful across sampling rates, its optimality as a split point may change because its relative proportion in the full spectrum varies. We therefore tested different thresholds:
>
> |Frequency threshold|16 kHz|24 kHz|
> |-|-|-|
> |without TLB|3.24|3.44|
> |2 kHz (TLB)|3.35|3.49|
> |3 kHz (TLB)|3.28|3.50|
> |4 kHz (TLB)|3.25|3.45|
>
> These results suggest that a frequency-aware split is consistently beneficial, and **2 kHz** is a strong default, although not always the absolute optimum (3 kHz is slightly better at 24 kHz).
>
> Regarding **languages**, we use English benchmark datasets mainly for fair comparison, as most baseline models are evaluated on English data. Our design is based on general acoustic properties of speech, especially low-frequency pitch and formant structures. We agree that spectral distributions vary across languages, so the optimal thresholds may differ. We will clarify this limitation in the revision.
>
> **Regarding question 1**
>
> Our goal is to maintain strong performance while preserving **efficiency** and **generalization**, rather than pursuing SOTA on a single dataset.DVPD-P is designed under a much smaller computational budget than heavy predictive SOTA models (**2.41G** vs **34.58G**).  Fig. 5 shows that DVPD-P generally exhibits stronger generalization than other predictive models. Table 4 shows removing **FANC** consistently degrades performance, indicating that its frequency-aware non-uniform compression is effective in practice. We therefore view **FANC** not as a module for maximizing single-benchmark predictive SOTA, but as an efficient acoustic-view encoder for the full **DVPD** framework.
>
> **Regarding question 2**
>
> We conducted an additional experiment on the VoiceBank+DEMAND and computed **WER** using the same **Whisper-tiny** model for all methods. DVPD reduces the WER of noisy speech to **6.7%**, outperforming **PGUSE (9.3%)** and **StoRM (10.1%)**. These results are consistent with the PESQ trend and further support that DVPD improves not only perceptual quality but also intelligibility.
>
> **Regarding question 3**
>
> To better compare with other non-uniform frequency compression methods, we selected the band-split (BS) module in BSRNN, which is a strong representative of frequency-dependent modeling,  We replaced the FANC encoder with the BS module under the same DVPD framework (without TLB) to ensure a fair comparison.
>
> |Model|PESQ|WV-MOS|
> |-|-|-|
> |DVPD (BS)|2.45|3.34|
> |DVPD|2.99|4.16|
>
> With the BS module, the performance becomes substantially worse. This result demonstrates that our current design, DVPD with FANC as an integrated framework, is more effective than using the BS-based frequency modeling approach.
>
> **Regarding question 4**
>
> For **TLB**, the table in **weakness 2** shows that DVPD is **not overly sensitive** to the exact threshold choice: all frequency-aware TLB variants outperform the version without TLB, and **2 kHz** remains a robust default.
>
> For **FANC**, we apologize that the compression scheme was not described clearly enough. Our default design uses **no compression for 0–2 kHz**, **2× compression for 2–4 kHz**, and **4× compression above 4 kHz**. The motivation is to preserve the most important low-frequency structures while still achieving substantial computational savings at higher frequencies. We further conducted an ablation on the low-frequency uncompressed range:
>
> |Hz range|Compression|PESQ|MACs(G)|
> |-|-|-|-|
> |0–1 kHz|0–1: ×1, 1–4: ×2, >4: ×4 |3.25|8.9|
> |**0–2 kHz**|0–2: ×1, 2–4: ×2, >4: ×4 |3.35|10.2G|
> |0–3 kHz|0–3: ×1, 3–4: ×2, >4: ×4 |3.39|11.4G|
>
> These results suggest that **0–2 kHz** is a practical tradeoff: **0–1 kHz** over-compresses important low-frequency information, while **0–3 kHz** brings only marginal gain at noticeably higher cost.

---

> > ### Author Rebuttal · Reviewer_QAde · 2026-04-03
> >
> > The concerns have been addressed. Considering the contribution of this paper, I will keep my original positive rating.

---

### Official Review · Reviewer_Pcdv · 2026-03-17

**Soundness:** 3
**Presentation:** 3
**Significance:** 2
**Originality:** 2
**Overall Recommendation:** 4
**Confidence:** 5

**Summary:**

The paper introduces DVPD, Dual View Predictive Diffusion model for speech enhancement. It includes FANC - an encoder designed to encode spectrograms in frequency dependent manner. It also includes a module for capturing features from visual perspective. Lastly, for inference a Training-free Lossless Boost (TLB) strategy is also used to refine generation quality without any additional fine-tuning. Evaluations on speech denoising and speech super resolution show that DVPD can provide superior performance while keeping the computational complexity in check.

**Compliance With Llm Reviewing Policy:**

Affirmed.

**Final Justification:**

Several of my concerns were addressed, keeping positive score as is.

**Key Questions For Authors:**

Please address the weaknesses pointed above.

**Limitations:**

No.
It may be okay for this paper to not explicitly outline negative societal impact.

**Strengths And Weaknesses:**

Strengths

--  Bringing an acoustic perspective for spectrograms into the learning process is a sound approach for diffusion based speech enhancement.
-- Inference time TLB appears to provide good improvements.
-- From Table 1, the computational complexity of DVPD is considerably better than several other approaches with which it is compared to.

Weaknesses
-- DVPD appears to be largely similar to PGUSE – a hybrid predictive-generative approach. This appears to show up in the performance as well, where without TLB the gain does not appear to be significant. This does bring into question the significance of acoustic perspective for spectrograms, FANC encoder etc. (though intuitively they do make sense)
-- There are quite a few hyperparameters in the approach. It would be good to provide some discussions around it.
-- The idea of designing frequency dependent architecture for speech enhancement is well established. So it makes sense to apply it here. -- However, it would have been nice to see existing approaches like BSRNN getting adopted in DVPD, and how that compares to FANC.
-- A general comment – While application of diffusion (hybrid or standalone) to SE is great to see, in terms of performance 2-3 years old predictive methods still outperform diffusion based methods. This questions what diffusion for SE is actually missing.

---

> ### Author Rebuttal · Authors · 2026-03-28
>
> We sincerely thank the reviewer for these thoughtful comments and helpful suggestions.
>
> **Regarding weakness 1**
>
> DVPD is indeed inspired by PGUSE, since hybrid predictive-generative frameworks are among the strongest current directions in speech enhancement. However, although both belong to the same hybrid family, their core difference lies in the **spectrogram modeling paradigm**. PGUSE treats the spectrogram as a homogeneous 2D input, whereas DVPD explicitly exploits its **acoustic-frequency structure** via **FANC** and **visual-texture structure** via **LISA**. Even **without TLB**, DVPD already outperforms PGUSE in Table 1 (PESQ **2.99 vs. 2.95**) while using only about **35% parameters** and **40% MACs**, showing a better quality-efficiency tradeoff. Table 4 further verifies the dual-view design: removing **LISA** causes the largest degradation (**-0.28 PESQ**), and removing **FANC** also leads to a clear drop (**-0.06**). Moreover, in Fig. 5, DVPD degrades much less than PGUSE under distribution shift (**0.08 vs. 0.19**), suggesting that the acoustic perspective is especially beneficial for robustness. Therefore, DVPD is not a simple variant of PGUSE, but a distinct lightweight design in which the acoustic/visual dual-view modeling yields consistent empirical benefits.
>
> **Regarding weakness 2**
>
> We agree and will clarify the hyperparameters more explicitly in the revision. They fall into three groups:
>
> (1) **Architecture-level parameters** in **FANC/LISA** are not free-form knobs, but are motivated by the non-uniform structure of speech spectrograms. In FANC, the **0–2 / 2–4 / >4 kHz** partition preserves low-frequency pitch/formant structures while progressively compressing higher frequencies. In LISA, dilations **3/5/7** provide a lightweight multi-scale receptive field for harmonic, formant, and transient patterns. The ablation **w/o FANC** confirms that this design is beneficial rather than arbitrary.
>
> (2) **Inference-time parameters** (**$\alpha$, $T_{rs}$, $N$**) are analyzed in Appendix E. In particular, **$\alpha=0.4$** gives the best tradeoff, and the studies on $T_{rs}$ and $N$ show that the defaults are empirically justified rather than aggressively tuned.
>
> (3) **TLB parameters** deserve clearer discussion. TLB is only applied to the diffusion branch and uses two parameter types: **$s$** (skip connections, for high-frequency compensation) and **$b$** (backbone, for residual-noise suppression and low-frequency refinement). Since we use a 3-stage U-Net but only modulate the first two stages, this yields **$s_1,s_2,b_1,b_2$**, each further split into low/high-frequency parts, resulting in **8 parameters** in total. We will reorganize this discussion and provide a clearer table in the revision.
>
> **Regarding weakness 3**
>
> **BSRNN** is a strong representative of frequency-dependent modeling. To directly test this, we replaced **FANC encoder and decoder** with the **band-split (BS) module and band-merge module** from BSRNN in the same DVPD framework without TLB:
>
> | Model | PESQ | WV-MOS |
> |-------|------|--------|
> | DVPD (BS) | 2.45 | 3.34 |
> | DVPD | 2.99 | 4.16 |
>
> The BSRNN-style BS module performs substantially worse in our framework. A plausible reason is that it is designed for a different architectural purpose, while **FANC** is tailored as a lightweight frequency-aware front-end for DVPD. This result supports that **FANC is a more suitable encoder for our efficiency-oriented hybrid design**.
>
> **Regarding weakness 4**
>
> We agree that on in-distribution benchmarks such as **VoiceBank+DEMAND**, strong predictive models (e.g., **MP-SENet**, PESQ **3.50**) still outperform diffusion-based methods (DVPD: **3.35**). This is expected, since predictive models excel at deterministic mappings under matched training/testing conditions. However, we believe the main value of diffusion-based SE is not peak performance on a single benchmark, but **robustness under distribution mismatch**. As shown in **Fig. 5**, when test data deviate from training conditions (e.g., **WSJ0-UNI → VBDMD / WSJ0-CE3**), DVPD shows only a small PESQ drop (**0.08–0.10**), while predictive models degrade much more (**0.25–0.31**). In real applications, such OOD robustness can be more important than the best in-distribution score. The above advantages and limitations are intrinsic to diffusion models, arising from the uncertainty modeling in the progressive iterative refinement process for spectrogram reconstruction.

---

> > ### Author Rebuttal · Reviewer_Pcdv · 2026-04-03
> >
> > Several of the concerns were adequately addressed. Some additional justifications for weaknesses 1 and 4 in particular would have made it better.

---

> > > ### Author Response · Authors · 2026-04-05
> > >
> > > For **Weakness 1**, we sincerely thank the reviewer for the opportunity to further clarify the distinctions between our method and PGUSE, as well as the necessity of our design. We provide the following detailed comparisons and analyses.
> > >
> > > **1. Structural Comparison with PGUSE**
> > >
> > > Our model is fundamentally built upon a **dual-view design**, which jointly models spectrograms from both an **acoustic view** and an **image view**, whereas PGUSE primarily focuses on the acoustic view.
> > >
> > > **Image view.** We introduce **LISA**, which explicitly treats the spectrogram as a 2D image and models its texture patterns. This enables the diffusion process to fully exploit its strength in iterative refinement of fine-grained structures. In contrast, PGUSE adopts DPRA, which processes features separately along time and frequency axes and does not explicitly model spectrogram textures from an image perspective.
> > >
> > > **Acoustic view.** Although both methods consider frequency-dependent modeling, our **FANC** encoder/decoder employs a **non-uniform three-band decomposition** (low/mid/high frequencies with no/moderate/strong compression), which better matches the non-uniform information density across frequency bands. Moreover, multi-scale dilated convolutions are incorporated within each band to enlarge the receptive field without increasing computational cost. In comparison, PGUSE only adopts a two-band (low/high) split and does not include multi-scale dilation.
> > >
> > > **Diffusion inference strategy.** In addition, we propose **TLB**, a training-free inference strategy tailored for U-Net-based diffusion models. PGUSE neither adopts a U-Net-based formulation nor includes such an inference mechanism.
> > >
> > > **2. Comparison in Performance and Complexity**
> > >
> > > We summarize the comparison between DVPD and PGUSE as follows:
> > >
> > > - **Params:** 1.9M (DVPD) vs. 5.1M (PGUSE)
> > > - **MACs:** 10.2G (DVPD) vs. 26.3G (PGUSE)
> > >
> > > These results demonstrate that DVPD is significantly more lightweight. For a fair comparison, we evaluate performance both with and without TLB under PESQ and DNSMOS metrics.
> > >
> > > | Model | PESQ | DNSMOS |
> > > |-|-|-|
> > > | PGUSE | 2.95 | 3.09 |
> > > | PGUSE + TLB | **3.02** | **3.55** |
> > > | DVPD | 2.99 | 3.47 |
> > > | DVPD + TLB | **3.13** | **3.63** |
> > >
> > > Regardless of whether TLB is applied, DVPD consistently outperforms PGUSE. This shows that our method achieves **better performance with significantly lower computational cost**.
> > >
> > > **3. Effectiveness of Key Modules**
> > >
> > > We further validate the necessity of each module via ablation studies:
> > >
> > > - **FANC**: Removing FANC results in a PESQ drop of 0.06.
> > > - **LISA**: Removing LISA leads to a significant PESQ drop of 0.28, demonstrating its critical role in modeling spectrogram textures.
> > > - **TLB**: Removing TLB causes a PESQ drop of 0.16.
> > >
> > > Among these components, **LISA** and **TLB** contribute most to performance gains, while **FANC**, although yielding moderate performance improvement, significantly reduces computational cost through non-uniform compression, supporting the lightweight design. Therefore, all modules play indispensable and non-replaceable roles.
> > >
> > > **Summary.** In summary, DVPD differs from PGUSE in architecture (dual-view design), key modules (FANC, LISA, TLB), and performance-efficiency trade-off. Each module plays a critical and non-replaceable role in maintaining these advantages.
> > >
> > > **Regarding the Weakness 4**
> > >
> > > We thank the reviewer for this insightful question. Beyond the general pros and cons discussed in our previous rebuttal, we note that VoiceBank+DEMAND is the **only** dataset where our diffusion-based model underperforms predictive methods. We attribute this gap to a mismatch between current diffusion paradigms and the requirements of speech enhancement (SE) in structured noise scenarios.
> > >
> > > **1. SE as a regression problem.** SE is essentially a **conditional regression task** that requires precise mapping from noisy input to its clean counterpart. Predictive models are well aligned with this objective, while diffusion models, being designed for conditional generation, cannot strictly enforce exact reconstruction. This limitation becomes more evident on structured datasets like VoiceBank+DEMAND.
> > >
> > >
> > > **2. Inefficiency and objective mismatch.** Diffusion models rely on multi-step refinement, which incurs iterative approximation errors and is unnecessary for structured noise. More importantly, diffusion models are inherently designed for perceptual generation rather than precise signal reconstruction. As a result, they do not explicitly optimize for sample-level fidelity, leading to a mismatch with distortion-oriented objectives, and consequently suboptimal performance under such metrics.
> > >
> > > **Summary.** In summary, these limitations mainly appear in structured noise scenarios such as VoiceBank+DEMAND, where precise regression is more critical than generative flexibility. In this case, diffusion models lack strong conditional control and efficient, task-aligned inference, explaining the performance gap.

---

### Decision · Program_Chairs · 2026-04-30

**Decision:**

Accept (regular)

**Comment:**

This paper introduces DVPD, a lightweight Dual-View Predictive Diffusion model for speech enhancement that processes spectrograms as both acoustic frequency structures and visual textures. The reviewers agreed on the paper's strengths, resulting in positive scores (Weak Accepts) across the board. Reviewers specifically praised the model's exceptional efficiency-quality trade-off—achieving strong performance with significantly fewer parameters and MACs than existing models like PGUSE—and highlighted the architectural effectiveness of the FANC and LISA modules. During the review process, reviewers raised concerns regarding the model's similarity to prior architectures, the practical deployability and generalizability of the reference-dependent TLB strategy, missing RTF measurements, and performance gaps compared to heavy predictive SOTA models on specific datasets. In a thorough rebuttal, the authors successfully mitigated these issues by clarifying the distinct advantages of their dual-view design, demonstrating that the TLB strategy can be practically deployed using reference-free DNSMOS tiering, providing concrete RTF and WER, and explaining that the model is intentionally optimized for extreme efficiency and out-of-distribution robustness rather than peak in-distribution predictive performance. Given the compelling efficiency gains, solid technical execution, and the authors' effective resolution of the primary critiques, this paper presents a valuable contribution to the field and is recommended for acceptance.